# Visceral organ morphogenesis via calcium-patterned muscle constrictions

**Noah P Mitchell[1,2], Dillon J Cislo[2], Suraj Shankar[1,3], Yuzheng Lin[2], Boris I Shraiman[1], Sebastian J Streichan[2,4]***

[1]Kavli Institute for Theoretical Physics, University of California, Santa Barbara, Santa Barbara, United States; [2]Department of Physics, University of California, Santa Barbara, Santa Barbara, United States; [3]Department of Physics, Harvard University, Cambridge, United States; [4]Biomolecular Science and Engineering, University of California, Santa Barbara, Santa Barbara, United States

**Abstract** Organ architecture is often composed of multiple laminar tissues arranged in concentric layers. During morphogenesis, the initial geometry of visceral organs undergoes a sequence of folding, adopting a complex shape that is vital for function. Genetic signals are known to impact form, yet the dynamic and mechanical interplay of tissue layers giving rise to organs' complex shapes remains elusive. Here, we trace the dynamics and mechanical interactions of a developing visceral organ across tissue layers, from subcellular to organ scale in vivo. Combining deep tissue light-sheet microscopy for in toto live visualization with a novel computational framework for multi-layer analysis of evolving complex shapes, we find a dynamic mechanism for organ folding using the embryonic midgut of *Drosophila* as a model visceral organ. Hox genes, known regulators of organ shape, control the emergence of high-frequency calcium pulses. Spatiotemporally patterned calcium pulses trigger muscle contractions via myosin light chain kinase. Muscle contractions, in turn, induce cell shape change in the adjacent tissue layer. This cell shape change collectively drives a convergent extension pattern. Through tissue incompressibility and initial organ geometry, this in-plane shape change is linked to out-of-plane organ folding. Our analysis follows tissue dynamics during organ shape change in vivo, tracing organ-scale folding to a high-frequency molecular mechanism. These findings offer a mechanical route for gene expression to induce organ shape change: genetic patterning in one layer triggers a physical process in the adjacent layer – revealing post-translational mechanisms that govern shape change.

*For correspondence:
streicha@ucsb.edu

Competing interest: The authors declare that no competing interests exist.

## Editor's evaluation

Using state-of-the-art light sheet microscopy and novel image analyses, this study shows how genetically encoded contractions of the visceral muscle shape morphogenesis of the *Drosophila* midgut via mechanical coupling. Muscle contractions induce cell shape changes that cause convergence and extension of the gut epithelium, which folds due to its incompressible nature. Thus, this work defines novel mechanisms of convergence and extension and tissue folding via the interaction of a genetic program with physical processes.

## Introduction

Visceral organ morphogenesis proceeds by the assembly of layered cell sheets into tubes, which develop into complex shapes (*Nelson and Gleghorn, 2012*). Through this process, genetic patterning instructs cellular behaviors, which in turn direct deformations in interacting tissue layers to sculpt organ-scale shape. This motif arises, for instance, in the coiled chambers of the heart, contortions of

the gut tube, and branching airways of the lung (*Savin et al., 2011*; *Le Garrec et al., 2017*; *Metzger et al., 2008*). Meanwhile, elastic bilayer sheets highlight the potential for mechanical interactions alone to generate nontrivial 3D shape transformations (*van Rees et al., 2017*). While studies of monolayer tissue development in planar geometries imaged near the embryo surface or ex vivo have uncovered general principles (*Irvine and Wieschaus, 1994*; *Streichan et al., 2018*; *Saadaoui et al., 2020*; *Rauskolb et al., 2014*), following the dynamics of fully 3D visceral organ shape change has remained largely out of reach (*Nerurkar et al., 2019*; *Pérez-González et al., 2021*). Physical models inferred from static snapshots of organ morphology have proven useful in this regard, but connecting dynamics at the cellular and sub-cellular level with the dynamics of shape change at the organ scale through live imaging remains a new frontier (*Vignes et al., 2022*; *Savin et al., 2011*; *Shyer et al., 2013*).

Uncovering cell and tissue dynamics in a shape-shifting organ presents several challenges. A conceptual challenge is that visceral organs exhibit both genetic and mechanical interactions between multiple tissue layers (*Sivakumar and Kurpios, 2018*; *Huycke et al., 2019*). Technical challenges arise as well, since their complex, dynamic shapes develop deep inside embryos. Capturing dynamics in vivo therefore requires imaging methods that overcome image degradation due to scatter (*de Medeiros et al., 2015*) and a computational framework for analysis of complex shapes.

The embryonic midgut – composed of muscle cells ensheathing an endodermal layer, linked by extracellular matrix (*Figure 1A*) – offers a system in which we can overcome these challenges and probe in toto organ dynamics across tissue layers at sub-cellular resolution. Its size and the molecular toolkit of the model system render the midgut ideal for light-sheet microscopy (*Krzic et al., 2012*), tissue cartography (*Heemskerk and Streichan, 2015*), and non-neuronal optogenetics (*Guglielmi et al., 2015*). Hox genes expressed in the muscle layer are required for the midgut to form its four chambers, but the mechanism by which genetic expression patterns are translated into tissue deformation, and in turn to organ shape, remains unclear (*Figure 1B–C*; *Bienz, 1996*; *Hoppler and Bienz, 1994*; *Wolfstetter et al., 2009*; *Bienz and Tremml, 1988*; *Immerglück et al., 1990*; *Reuter and Scott, 1990*; *Panganiban et al., 1990*). Here, we connect this genetic patterning to mechanical interactions between layers during development and track the kinematic mechanism linking mechanical action to organ shape transformations. We find that dynamic, high-frequency calcium pulses drive patterned muscle contraction, inducing convergent extension in the endoderm to sculpt stereotyped folds.

## Results
### Live deep tissue imaging reveals bilayer morphogenesis

The midgut is a closed tube by stage 15 of embryonic development, residing 20–60 µm below the embryo surface (*Bate and Martinez Arias, 1993*). The organ first constricts halfway along its length, then constricts again to subdivide into four chambers (*Video 1*). Within 75–90 min after the onset of the first fold, the constrictions are fully formed, and the organ beings to adopt a contorted shape.

Quantitative characterization of these dynamics requires extraction of the full organ's geometry, which is challenging due to tissue scatter. We overcome this challenge by in toto live imaging using confocal multi-view light sheet microscopy (*de Medeiros et al., 2015*). In conjunction, we express tissue-specific markers using the GAL4-UAS system (*Brand and Perrimon, 1993*) in *klarsicht* embryos (*Welte et al., 1998*), which exhibit genetically-induced optical clearing (see Materials and methods). To translate this volumetric data into dynamics of the midgut surface, we combine machine learning (*Berg et al., 2019*) with computer vision techniques (*Márquez-Neila et al., 2014*; *Chan and Vese, 2001*) using an analysis package dubbed 'TubULAR' (*Mitchell and Cislo, 2022*). In this way, we are able to resolve sub-cellular structures with 1-min temporal resolution (*Figure 1D*, *Figure 1—figure supplement 1*, *Figure 1—figure supplement 2*, and *Video 2*).

We find that gut morphogenesis is stereotyped and exhibits reproducible stages (*Figure 1E*). The surface area grows by ~30% during folding (stages 15-16a) and remains constant by the time constrictions are fully formed (16b), despite continued shape change. The enclosed volume within the midgut decreases only gradually during this process, while the effective length of the organ – the length along its curving centerline – triples (*Figure 1E*, Materials and methods).

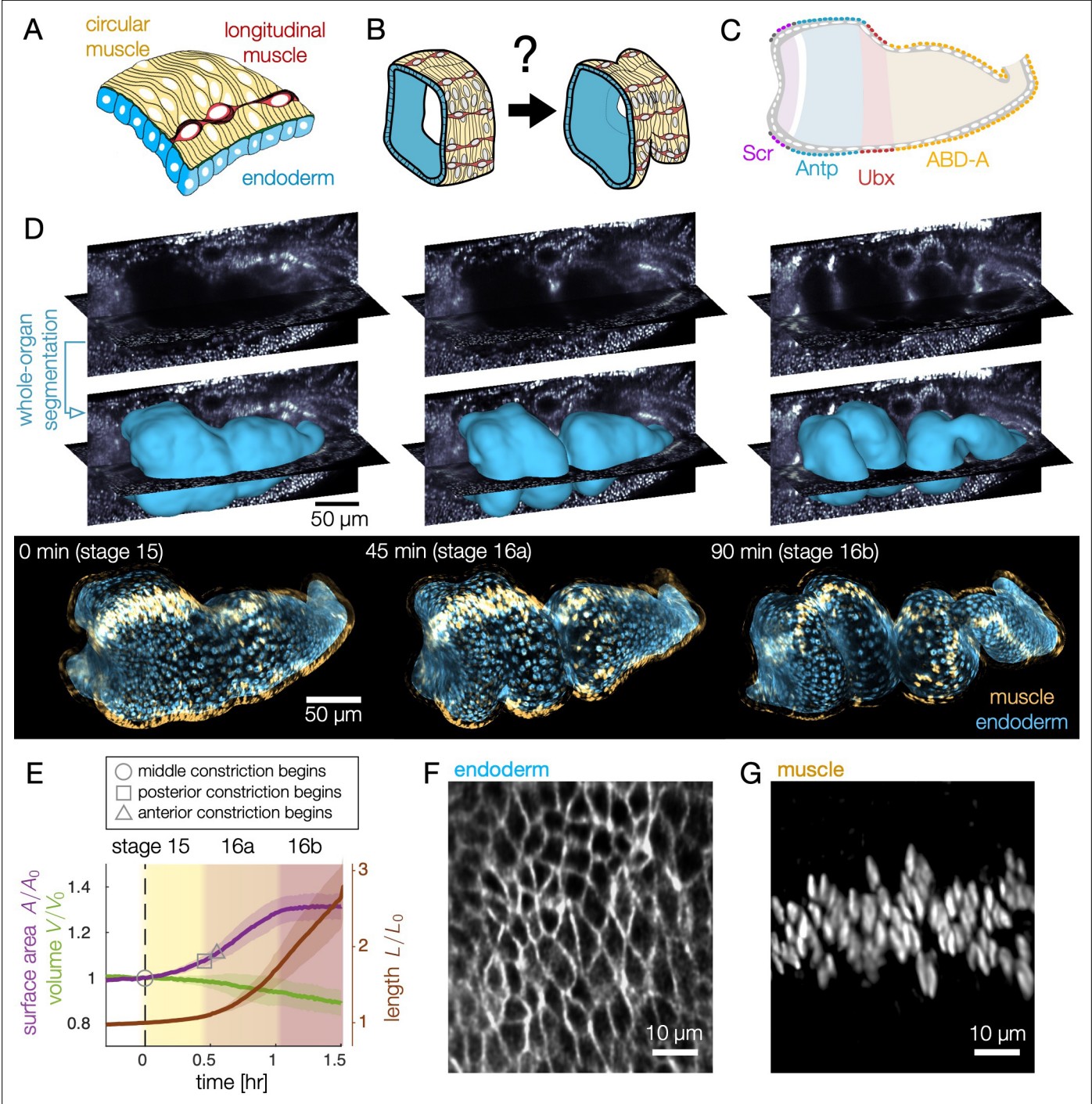

**Figure 1.** Deep tissue live imaging reveals bilayer gut morphogenesis. (**A–B**) Muscle and endoderm layers compose the midgut and interact to generate 3D shape. (**C**) Genetic patterning of hox transcription factors that govern midgut morphogenesis appears in the circumferential muscles. (**D**) Automatic segmentation using morphological snakes level sets enables layer-specific imaging, here shown for muscle (yellow) and endoderm (blue) for a *w,Hand>GAL4;UAS-Hand:GFP;hist:GFP* embryo. Morphogenesis proceeds first with a constriction cleaving the gut into two chambers (stage 15). Two more constrictions form a total of four chambers (16a) before the gut begins to coil (16b onward). Stages follow (***Campos-Ortega and Hartenstein, 1997***). (**E**) Surface area of the apical surface increases gradually during constrictions, but levels off by stage 16b. The enclosed volume decreases gradually, while the effective length of the organ – computed via the length of a centerline – nearly triples. Solid curves denote the means and shaded bands denote standard deviations ($N = 7$). Developmental timelines are aligned by the onset of the middle constriction (dashed black line). (**F–G**) Segmentation of the endoderm layer from MuVi SPIM imaging resolves individual cells, both in the endoderm and muscle layers, shown here at stage 15.

*Figure 1 continued on next page*

*Figure 1 continued*

The online version of this article includes the following figure supplement(s) for figure 1:

**Figure supplement 1.** Multi-view light sheet microscopy enables volumetric imaging.

**Figure supplement 2.** Organ segmentation via morphological snakes level sets enables layer-specific imaging and analysis.

**Figure supplement 3.** A centerline measures an effective length of the organ.

## Endodermal cell shape change underlies tissue shape change

How does this 3D shape change occur at the tissue and cellular scale? We first analyzed the endoderm layer. Inspection of these cells reveals strikingly anisotropic cell shapes before constrictions begin (*Figures 1F and 2A–B*). In order to quantify cell shape on this dynamic surface, we cartographically project into the plane using TubULAR (*Mitchell and Cislo, 2022*; *Aigerman and Lipman, 2015*). This projection defines a global coordinate system in which we unambiguously define the anterior-posterior (AP) and circumferential axes for all time points, even when the organ exhibits deep folds and contortions (*Figure 2—figure supplement 1* and *Mitchell and Cislo, 2022*).

By segmenting cell shapes, we find that endodermal cells are strongly anisotropic, with an average aspect ratio $a/b > 2$, and are globally aligned along the circumferential axis (*Figure 2A-C*). As constrictions develop, cells loose this anisotropy and even become elongated along the AP axis in posterior regions (*Figure 2B, C*, *Figure 2—figure supplement 2A*). Measurement of endodermal cell orientations reveal that this effect is not due to rotations (*Figure 2D*). As shown in *Figure 2—figure supplement 2A*, the initial anisotropy is patterned along the AP axis so that cells near two of the constriction locations are most anisotropic. While we do not presently know the mechanism of this patterning, it suggests these tissue regions may be primed for deformation by positional information before constrictions begin. Subsequent cell shape change is greatest near each constriction, as shown in *Figure 2—figure supplement 2B*.

Despite the large changes in aspect ratio, cell areas in the endoderm change only gradually (*Figure 2E*), such that cells converge along the circumferential axis while extending along the folding longitudinal axis. On a larger scale, the observed cellular deformation would collectively generate tissue movement called convergent extension. At the same time, other processes – such as oriented divisions or cell intercalations – could also contribute or counteract tissue-scale convergent extension (*Etournay et al., 2015*). However, we find no signs of cell division during this process, confirming previous observations (*Campos-Ortega and Hartenstein, 1997*). Moreover, although tracking quartets of cells in the anterior midgut revealed widespread intercalations (also called T1 transitions), the orientations of these events were not significantly biased for the early stages of constriction (*Figure 2—figure supplement 3* and *Video 3*).

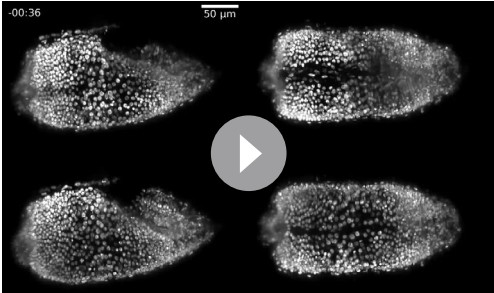

**Video 1.** Live imaging using confocal MuVi SPIM shows the shape change of the midgut as it folds into a coil of compartments. Here, we display panoramic views of fluorescently-labeled nuclei in both endoderm and muscle tissue layers of the midgut in a *w;48Y-GAL4;klar × w;UAS-histone::RFP* embryo. Maximum intensity projections of half-volumes from left lateral (upper left), right lateral (lower left), dorsal (upper right), and ventral views (lower right) exhibit the nearly isotropic resolution of our imaging setup.

https://elifesciences.org/articles/77355/figures#video1

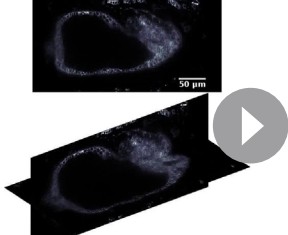

**Video 2.** Segmentation using computer vision techniques enables layer-specific imaging of midgut morphogenesis, highlighted here for a *w;48Y-GAL4;klar × w;UAS-CAAX::mCh* embryo. First, a morphological snakes level set identifies the midgut endoderm at a timepoint before the onset of constrictions. This surface follows the evolving shape of the organ during the subsequent dynamics. Rendering the data that intersects this dynamic surface allows us to visualize cells.

https://elifesciences.org/articles/77355/figures#video2

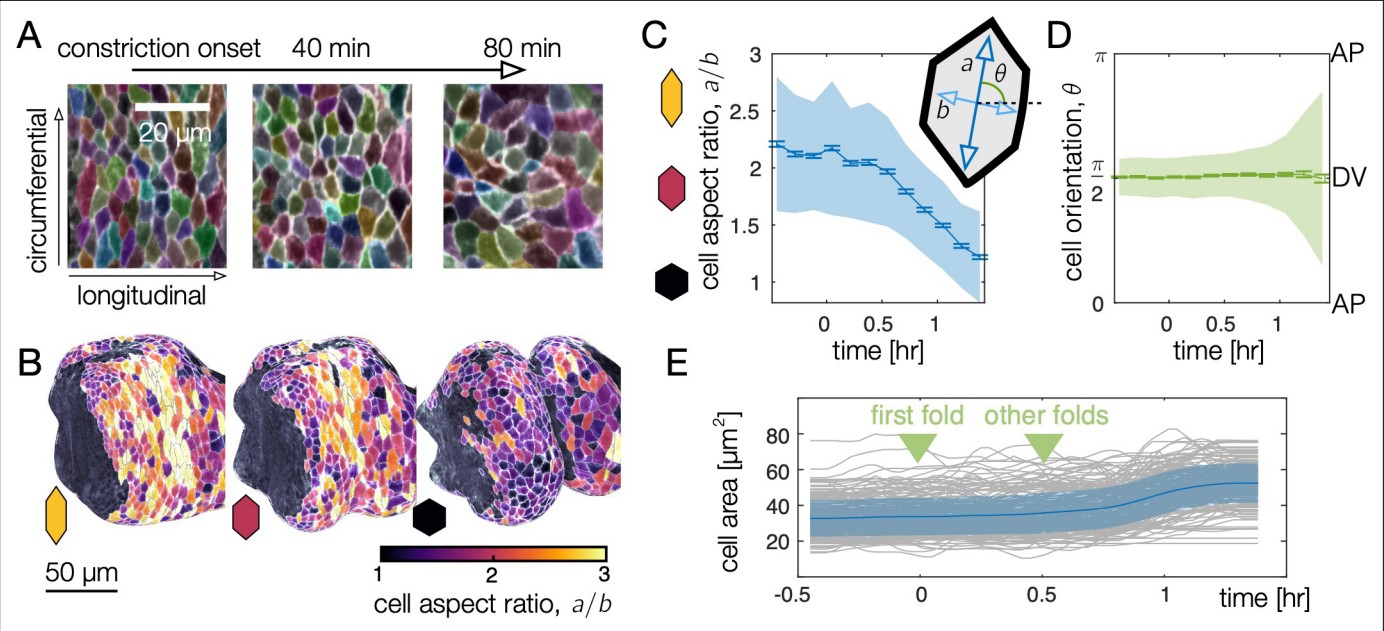

**Figure 2.** Endodermal cell shape changes underlie organ shape change. (**A**) Cell segmentation in a computationally flattened coordinate patch shows endodermal cells are initially elongated along the circumferential direction but change their shape during organ folding. (**B–C**) Cell aspect ratios evolve from $a/b > 2$ to $a/b \approx 1$, shown in 3D for cells near the anterior fold. (**C**) The mean endodermal cell aspect ratio averaged over the organ decreases during constrictions. Colored bands denote area-weighted standard deviations for 600–1300 segmented cells per timepoint, and tick marks denote standard error on the mean. (**D**) As cells change their aspect ratio, their orientations do not rotate. (**E**) Single-cell tracking shows gentle increase of cell areas through violent folding events, suggesting that cell area changes do not drive organ shape change. The blue curve and shaded region denote mean and standard deviation, with raw traces in gray.

The online version of this article includes the following figure supplement(s) for figure 2:

**Figure supplement 1.** Surface projection aids in quantifying cell shape change in the 3D tissue surface.

**Figure supplement 2.** Endodermal cells are initially most elongated near anterior and middle constrictions, and cell shape change is greatest near constrictions.

**Figure supplement 3.** Topological rearrangements occur in the endoderm but are not aligned during the earliest stages of gut constrictions, consistent with cell shape change being the dominant contributor to tissue deformation at the onset of constrictions.

This suggests that anisotropic cell shape change may be the primary contributor to tissue-scale shape change. We next tested this hypothesis, asking how in-plane, cell-scale shape change connects to out-of-plane, tissue-scale deformations during constrictions.

## Tissue-scale convergent extension via constriction

To understand the kinematic mechanism underlying organ shape, we must bridge spatial scales from cell deformation to meso-scale tissue deformation. Given that the midgut tissue is thin compared to the organ radius, cells exert forces on one another primarily through in-plane interactions, but in-plane mechanical stress can couple to curvature to generate out-of-plane motion (*Arroyo and Desimone, 2009*). In a nearly incompressible tissue constricting out-of-plane, cells do not change area, but may change shape, collectively driving in-plane motion. In particular, the out-of-plane motion $v_n$ would couple to in-plane motion $\mathbf{v}_{\parallel}$ through the mean curvature $H$, such that

$$2Hv_n \approx \nabla \cdot \mathbf{v}_{\parallel}, \tag{1}$$

where $\nabla \cdot \mathbf{v}_{\parallel}$ is the in-plane divergence of the tissue motion along the surface. Such a kinematic constraint guides the shape changes that result from prescribed patterns of mechanical stresses in the tissue.

We hypothesized that the constricting midgut may behave as nearly incompressible, given that we found cell areas to vary only gradually during constrictions. To test this, we extract whole-organ tissue deformation patterns and find strong out-of-plane motion and in-plane dilatational flows concentrated

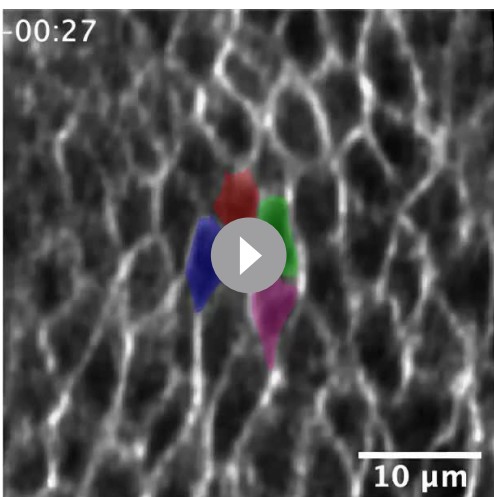

**Video 3.** An example tissue patch shows that cell rearrangements in the endodermal layer (i.e 'T1 transitions' or 'intercalation events') occur during gut morphogenesis. Here, the blue and green cells are not initially neighbors, but rearrange to separate the red and pink cells. Subsequently, the blue and pink cells lose their connection as well, as do the green and pink cells. Each frame shows the tissue patch mapped to 2D by a rigid flattening of the surface in a manner that the map is nearly conformal near the center of the image, with metric components $g_{11} \approx g_{22} \approx 1$. T1 transitions are not biased in their orientation until late stages of morphogenesis, consistent with cell shape change being the dominant contributor to tissue shear.

https://elifesciences.org/articles/77355/figures#video3

near folds (*Figure 3A*, *Video 4*, and *Figure 3—figure supplement 1*). Remarkably, we find that the pattern of out-of-plane deformation almost entirely accounts for in-plane dilatational motion in the gut, with only a small change in local tissue areas (*Video 5*). This tight link suggests that the tissue behaves as an incompressible medium. As shown in *Figure 3B*, these terms match with 97% correlation, leaving a residual in-plane growth residue at the level of ~1% per minute. This slow residual area growth, which is primarily concentrated in the lobes of rounding gut chambers, accounts for the surface area growth noted in *Figure 1E* and the cellular area growth in *Figure 2E*.

Because the organ is curved into a tube, constrictions converge the tissue along the circumferential axis, and tissue incompressibility couples inward motion to extension along the longitudinal axis to preserve areas (*Figure 3C*). We dub this kinematic mechanism 'convergent extension via constriction': as the tissue constricts with an inward normal velocity, the length of the tissue along the circumferential direction shortens while curves along the longitudinal (AP) axis of the organ lengthen, keeping the areas of cells approximately constant (*Figure 3C, F*, *Figure 3—figure supplement 2*, *Figure 3—figure supplement 3*). As a consequence of tissue incompressibility and localized constrictions, the resulting area-preserving deformations are largest near constrictions (*Figure 3—figure supplement 4*), mirroring the pattern of cell-scale deformations. Although the shape of the organ becomes increasingly complex, in-plane deformations remain globally aligned in the material coordinate system: the tissue converges and extends along the circumferential and longitudinal axes, respectively, even as these axes deform in 3D space as morphogenesis proceeds (*Figure 3—figure supplement 4*).

Finally, we find that tissue convergent extension is accounted for primarily by our previous measurement of cell shape change. Since the early stages of midgut constrictions have no divisions or oriented cell intercalations, we hypothesized that cell shape change alone can explain the tissue scale convergent extension. *Figure 3—figure supplement 5* shows a quantitative match between cell shape changes and tissue convergent extension, indicating that local cell shape changes primarily mediate tissue-scale convergent extension during the early stages of constrictions. Later during stage 16, the tight association between cell-scale and tissue-scale deformations loosens, corresponding to contributions from cell intercalations (*Blanchard et al., 2009*).

In short, we established a link from endodermal cell shape change to tissue-scale folding – in which incompressibility couples out-of-plane deformation to in-plane motion – resulting in convergent extension via constriction (*Figure 3G*). What mechanical process drives strong, localized contractions at the folds?

## Muscle contractions drive cell and tissue shape change

It is known that embryos with either disrupted muscle or endoderm structure fail to fold (*Bilder and Scott, 1995*; *Singer et al., 1996*; *Wolfstetter et al., 2009*; *Wolfstetter et al., 2017*), as do embryos lacking integrins linking the two layers (*Devenport and Brown, 2004*). This suggests that gut morphogenesis requires an interaction between muscle and endodermal layers. At the same time, hox genes – which are expressed exclusively in the muscle layer – have been linked to the successful

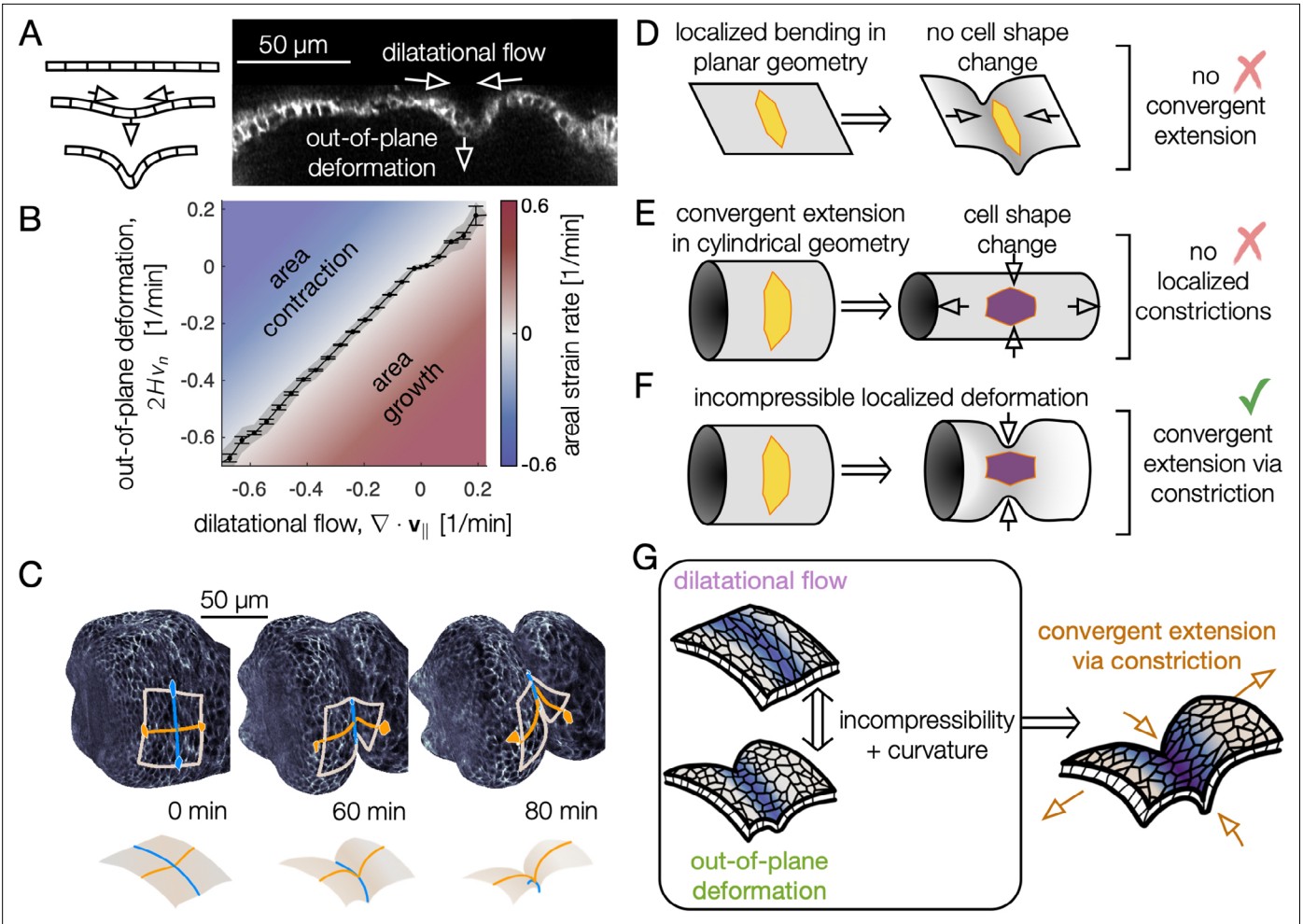

**Figure 3.** Incompressible tissue dynamics reveal convergent extension via constrictions. (**A**) Localized constrictions couple dilatational in-plane velocity patterns to out-of-plane deformation near folds. (**B**) In-plane divergence and out-of-plane deformation are correlated at the 97% level, signaling nearly incompressible behavior ($N = 3$ embryos, with kinematics sampled in 320 non-overlapping tissue patches per minute for 0<90 min). Gray band denotes standard deviation and ticks denote standard error on the mean for each bin. Here, $H$ denotes mean curvature, $v_n$ is the normal (out-of-plane) velocity, and $\nabla \cdot \mathbf{v}_{\parallel}$ is the covariant divergence of the in-plane velocity. (**C**) The tissue converges along the circumferential direction as cells sink into the constriction (blue) and extends along the bending longitudinal profile (orange) to preserve the area of a tissue patch. (**D**) In contrast to the curved gut, localized bending of a flat, incompressible sheet requires no cell shape change, and thus no tissue-scale convergent extension. (**E**) Cell shape deformations converging along the circumferential axis and extending along the the AP axis would generate tissue convergent extension corresponding to uniform constriction of a tube, but no localized constrictions would form. (**F**) Localized constriction of an incompressible sheet exhibits cell shape change without cell area change in the absence of oriented divisions or oriented cell intercalations. The cell shape extends along the bending longitudinal (AP) axis. (**G**) Convergent extension via constriction follows as a geometric consequence of localized constrictions of the tubular organ without local area change.

The online version of this article includes the following figure supplement(s) for figure 3:

**Figure supplement 1.** Dilatational flow patterns are tightly coupled to out-of-plane deformation throughout midgut constrictions, indicating a nearly incompressible tissue behavior.

**Figure supplement 2.** Convergent extension via constriction links in-plane tissue shape change with out-of-plane deformation.

**Figure supplement 3.** Three-component decomposition of kinematics elucidates coupling between compressibility and convergent extension.

**Figure supplement 4.** Tissue shear generates 3D convergent extension during constrictions, as captured in the Beltrami coefficient – a local measure of anisotropic, area-preserving deformation.

**Figure supplement 5.** Cell shape change quantitatively accounts for tissue-scale convergent extension during early stages of midgut constrictions.

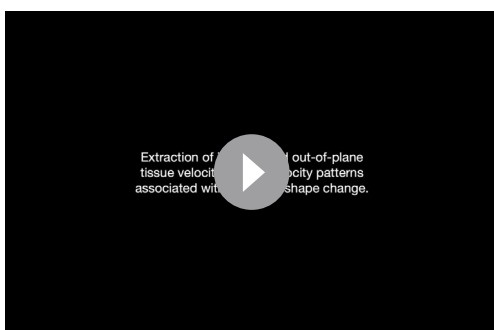

**Video 4.** Extracted in-plane and out-of-plane tissue velocities show signatures associated with stages of shape change. The tangential (in-plane) component of the velocity ($\mathbf{v}_{\parallel}$, top panel) is colored by its orientation in the $(s, \phi)$ pullback plane described in the Materials and methods. In this 'unwrapped' pullback plane, ventral tissue occupies the center of the image, anterior is left, and posterior is right. For example, purple regions flow towards the midgut's posterior and orange regions flow towards the anterior. Opacity of the colored signal is proportional to the magnitude of the tissue velocity. The normal (out-of-plane) component of the velocity ($v_n$, bottom panel) is red for motion in the endoderm's apical direction (toward the inside of the gut) and blue for motion in the basal (outer) direction. The corresponding midgut surface is shown on the left for all timepoints.

https://elifesciences.org/articles/77355/figures#video4

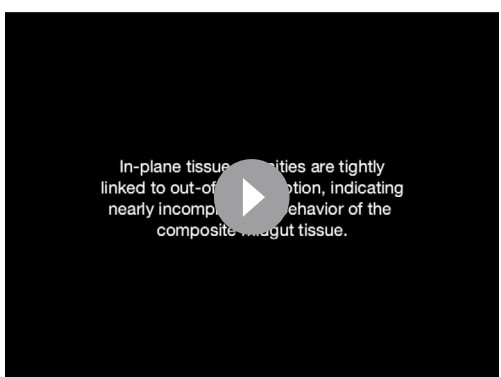

**Video 5.** In-plane tissue velocities are tightly linked to out-of-plane motion, indicating nearly incompressible behavior of the composite midgut tissue. For a representative embryo, we plot the out-of-plane deformation $2Hv_n$ on the left, both as a left lateral view in 3D and in the $(s, \phi)$ pullback plane. Here, $v_n$ is the normal velocity and $H$ is the mean curvature, which is positive for a cylinder but negative for sharp folds whose outer radius of curvature is smaller than their inner circumference. In the middle column, we plot the dilatational flow for this embryo, defined as the covariant divergence of the in-plane velocity, $\nabla \cdot \mathbf{v}_{\parallel}$. In the right column, we plot the difference between these two quantities, which measures the rate of isotropic expansion or contraction of the tissue.

https://elifesciences.org/articles/77355/figures#video5

formation of specific folds (*Figure 4A–B*; *Tremml and Bienz, 1989*). In particular, *Antp* mutants lack the anterior constriction lying near the center of the Antp domain (*Figure 4C*), while *Ubx* mutants lack the middle constriction lying at the posterior edge of the Ubx domain (*Figure 4D*). In this system, genetic patterning of the endoderm occurs via genetic patterning from the muscle layer (*Bienz, 1996*; *Mendoza-Garcia et al., 2021*), so it is possible that constrictions result from a *genetic* induction process. Alternatively, *mechanical* interactions between the layers could induce a program of convergent extension in the endoderm – with patterned deformation of the muscle layer sculpting a passive, tethered endoderm (*Reuter and Scott, 1990*) or triggering active endodermal cell shape change.

To clarify the relationship between layers during constriction dynamics, we first measured relative motion of the muscle layer against the endoderm. By tracking both circumferential muscle nuclei and endoderm nuclei in the same embryo, we find that these two layers move together, with initially close nuclei separating by ~5 µm per hour (*Figure 4E–F*, *Video 6*, *Figure 4—figure supplement 1*, and *Figure 4—figure supplement 2*). This result is consistent with the notion that the two layers are tightly tethered by the integrins and extracellular matrix binding the heterologous layers (*Devenport and Brown, 2004*).

Based on this tight coupling, we hypothesized that muscle mechanically induces shape change in the tethered endoderm. To test this hypothesis, we inhibited contractility of the muscle layer by driving *UAS-CIBN UAS-CRY2-OCRL*, a two-component optogenetic construct that recruits *OCRL* to the plasma membrane to dephosphorylate PI(4,5)P$_2$. This process has been shown to abolish actomyosin contractility in other developmental contexts by releasing actin from the plasma membrane (*Guglielmi et al., 2015*). Driving *CRY2-OCRL* with *Antp-GAL4* under continuous activation of blue light reliably prevented anterior constrictions (*Figure 4G*). Likewise, driving *CIBN* and *CRY2-OCRL* under continuous blue light activation in muscle regions posterior to the anterior fold using *Ubx-GAL4 M1* locally inhibited constriction dynamics. We note that *Ubx-GAL4 M1* embryos express *Ubx* in a larger domain than the endogenous WT Ubx domain due to differences in its regulation (*Garaulet et al., 2008*), but *Ubx-GAL4 M1* embryos nonetheless execute all three constrictions in the absence of *UAS-CRY2-OCRL* under similar imaging conditions. Inhibiting

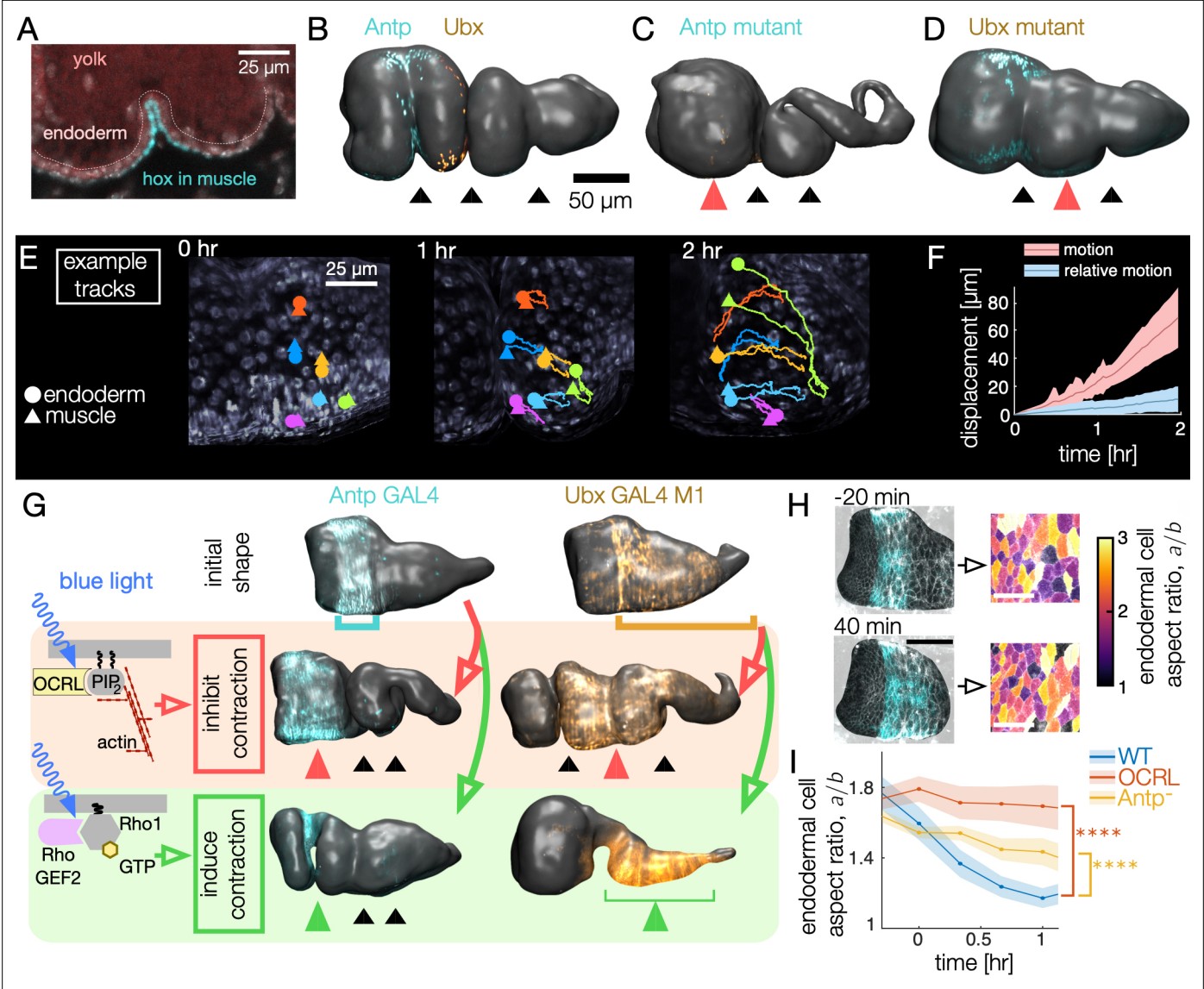

**Figure 4.** Muscle contractions mechanically couple to the endoderm layer, inducing cell shape change and convergent extension. (**A–B**) Hox genes *Antp* and *Ubx* are expressed in the circumferential muscle in discrete regions, shown for surfaces extracted from light-sheet imaging. (**C–D**) Hox genes control organ shape: *Antp* and *Ubx* mutants lack anterior and middle constrictions, respectively. (**E**) Muscle and endoderm layers move together. By computationally extracting both muscle and endoderm layers in an embryo expressing both fluorescent circumferential muscle and endoderm (*w,Hand>GAL4;UAS-Hand:GFP;hist:GFP*), we track relative motion of initially close muscle-endoderm nuclei pairs. (**F**) Muscle-endoderm nuclei pairs show modest relative motion compared to the integrated motion of the tissue ($N = 81$ pairs, colored bands denote standard deviations). (**G**) Optogenetic inhibition of contractility via *CRY2-OCRL*, which dephosphorylates PI(4,5)P$_2$ (*Guglielmi et al., 2015*), mimics hox mutant behaviors ($N = 11$ each), and stimulation of muscle contraction via recruitment of *CRY2-RhoGEF2* to the plasma membrane (*Izquierdo et al., 2018*) drives ectopic folding ($N = 5$ each). (**H**) Inhibiting muscle contraction via *CRY2-OCRL* prevents endoderm cell shape change, shown for snapshots before and after the anterior constriction would normally form. The black scale bar is 50 μm, and white scale bar in images of segmented cells is 25 μm. (**I**) Measurements of endodermal cell anisotropy over time confirm that mechanical inhibition in the muscle reduces cell shape change in the endoderm (blue, $p = 1 \times 10^{-22}$). *Antp* mutants also exhibit reduced endoderm cell shape change, consistent with *Antp* regulating muscle contraction (yellow, $p = 4 \times 10^{-9}$). Each datapoint is the weighted average of multiple adjacent timepoints from two to three embryos with at least 30 cells per timepoint segmented in each embryo. Colored bands denote standard error on the mean, and **** denotes $p < 0.0001$.

The online version of this article includes the following figure supplement(s) for figure 4:

**Figure supplement 1.** Relative motion of endodermal and mesodermal nuclei is small compared to motion of the tissue.

**Figure supplement 2.** Motion of the muscle nuclei with respect to the endoderm is not coherent.

**Figure supplement 3.** Optogenetic knockdown of muscle contractility inhibits endodermal shape changes, mimicking mutant behavior.

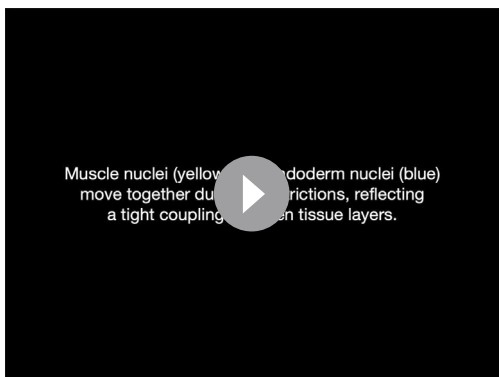

**Video 6.** Muscle nuclei (yellow) and endoderm nuclei (blue) move together during constrictions, reflecting a tight coupling between tissue layers. Here, we false color two selected layers of a *w,Hand>GAL4;UAS-Hand:GFP;hist:GFP* embryo in yellow (muscle layer) and blue (endoderm).
https://elifesciences.org/articles/77355/figures#video6

contraction in selected regions therefore mimics the genetic mutants known to remove folds.

Given that muscle contractility is required, we asked if optogenetically inducing actomyosin contraction in the muscle is sufficient to induce constrictions. Indeed, optogenetic activation using the *CIBN UAS-RhoGEF2* system in the Antp region generates an anterior fold on demand on the timescale of a few minutes, even if induced long before the constriction would normally begin (*Figure 4G*). Similarly, activation of the *Ubx-GAL4 M1* domain results in a nearly uniform constriction that dramatically alters the shape of the organ, forcing the yolk to flow into the anterior chamber. Additional optogenetic experiments inhibiting contractility of all muscles likewise led to folding defects ($N = 13$, *w;UAS-CIBN::pmGFP;Mef2-GAL4/UAS-mCherry::CRY2-OCRL*). We conclude that muscle contractility is necessary for constrictions and inducing contraction and the associated downstream behaviors is sufficient to generate folds.

We then asked how these macro-scale perturbations on organ shape are linked to cell shapes in the endoderm. In contrast to wild-type, optogenetic inhibition of muscle contractility significantly reduced endodermal cell shape change. As shown in *Figure 4H–I* and *Figure 4—figure supplement 3*, cell segmentation of the endoderm reveals nearly constant aspect ratios during optogenetic inhibition of muscle contraction in the Antp domain: the endoderm cells near the Antp domain undergo reduced convergent extension when muscle contraction is locally disrupted (single-sided z-test: $p = 1 \times 10^{-6}$ for difference after 1 hr, $p = 1 \times 10^{-22}$ for sustained difference between curves, see Methods). We also observe analogous reduction of endodermal cell shape change in *Antp* mutants, which lack anterior folds (*Figure 4I* and *Figure 4—figure supplement 3*, single-sided z-test: $p = 7 \times 10^{-3}$ for difference after 1 hr, $p = 4 \times 10^{-9}$ for sustained difference between curves). Thus, the endodermal program of convergent extension is induced by mechanical interaction with the contracting muscle layer.

## Calcium pulses spatiotemporally pattern muscle contractility

What mechanism triggers muscle contractions, allowing such sharp folds to arise? Recent studies have shown that calcium signaling triggers muscle contractions in a wide range of contexts (*Kuo and Ehrlich, 2015*). If hox genes use calcium signaling to pattern muscle contraction in the midgut, we would predict that calcium pulses should appear near localized constrictions. Furthermore, hox gene mutants lacking constrictions would not exhibit localized calcium pulses, and inhibition of the cell biological mechanism translating calcium into mechanical contraction should likewise inhibit constrictions.

To test for a link from hox genes to organ shape through this mechanism, we first imaged a fluorescent probe of calcium dynamics (GCaMP6s) in the muscle layer. As shown in *Figure 5A–E* and *Video 7*, dynamic calcium pulses appear in the muscle layer in regions localized near all three midgut constrictions. Additionally, these calcium pulses are patterned in time, appearing only at the onset of constriction for each fold (*Figure 5—figure supplement 1*).

To test whether hox genes pattern shape change through calcium dynamics, we measured GCaMP6s activity in flies mutant for *Antp* that lack an anterior constriction. As shown in *Figure 5F*, we found that calcium activity was almost entirely absent during stages 15–16. Calcium activity is strongly reduced at the location of the missing anterior constriction (single-sided z-test: $p = 2 \times 10^{-8}$) and subsequent calcium pulses are repressed within the vicinity of the region for the hour after the constriction would normally initiate (single-sided z-test: $p = 1 \times 10^{-13}$ within 50 μm, *Video 8* and *Figure 5—figure supplement 2*). In contrast, calcium pulses continue to appear at the anterior constriction in *Ubx* mutants, suggesting local control of calcium pulses by *Antp* in the anterior constriction (*Figure 5—figure supplement 3*). The hox gene *Antp* is therefore upstream of dynamic calcium pulses.

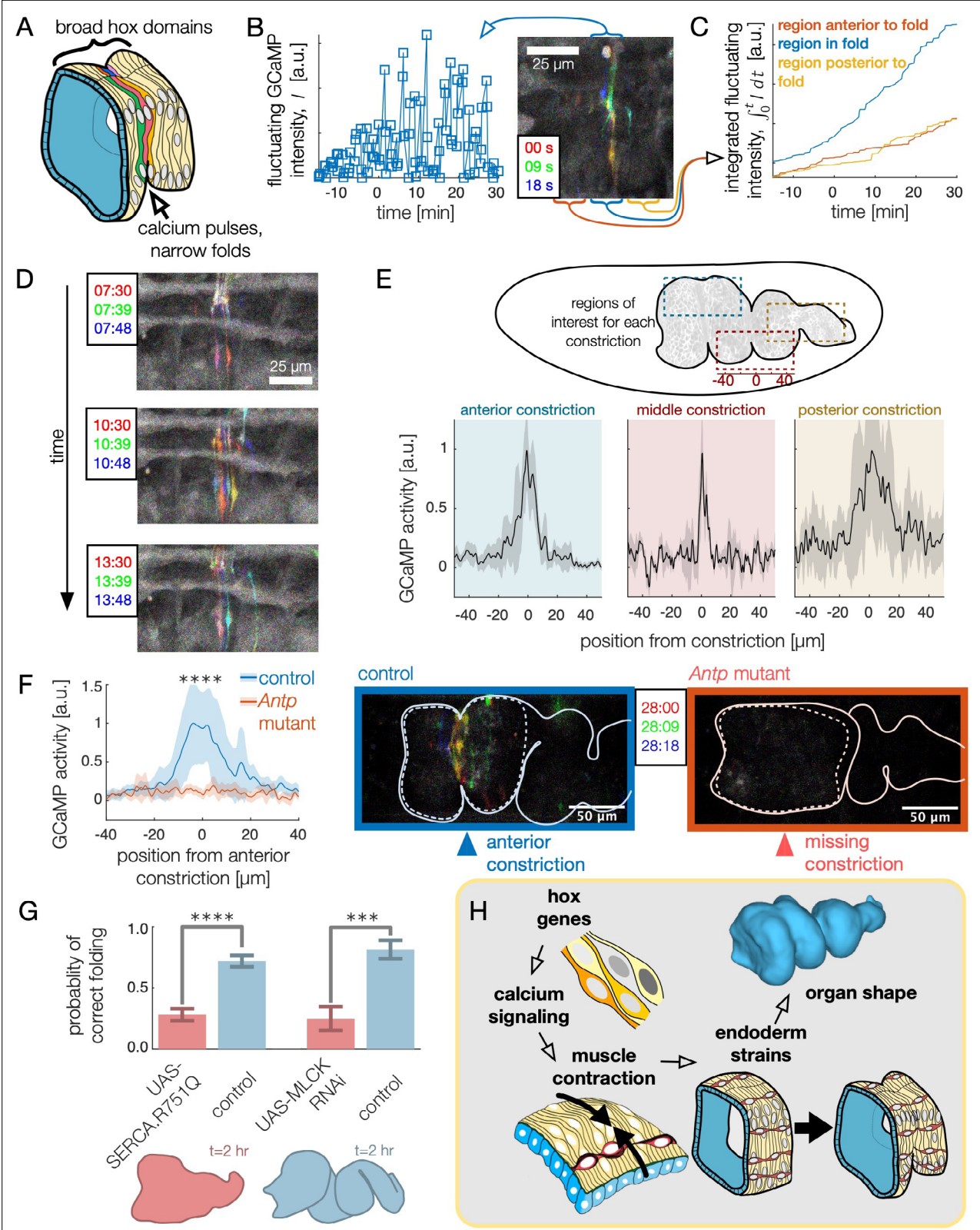

**Figure 5.** High frequency calcium pulses mediate muscle contraction, linking hox genes to organ shape through tissue mechanics. (**A**) Dynamic calcium pulses appear near the anterior fold, localized to a region more narrow than the Antp domain. (**B**) Transient pulses in GCaMP6s intensity occur on the timescale of seconds and increase in amplitude when folding begins ($t = 0$). Red, green, and blue channels of images represent maximum intensity projections of confocal stacks separated in time by 9 seconds, here and below. (**C**) Integrated transient pulses for the embryo in (**B**) show calcium pulses

*Figure 5 continued on next page*

*Figure 5 continued*

are localized near the fold: GCaMP6s signals 20 μm in front (red) or behind the fold (yellow) are less intense. (**D**) Snapshots of GCaMP6s fluorescence in muscle cells demonstrate calcium activity near constrictions. Each frame is a composite of three subsequent snapshots in red, green, and blue, so that transient pulses appear as colored signal, while background appears gray. Different muscle cells report calcium activity in adjacent frames. (**E**) Average fluorescent activity during the first 15 min of folding show localized signatures at each constriction, with particularly sharp peaks in the middle and anterior constrictions ($N = 5$, $N = 2$, and $N = 7$ for anterior, middle, and posterior folds, respectively). (**F**) In *Antp* mutants, GCaMP6s fluorescence is significantly reduced ($p = 2 \times 10^{-8}$) and is not localized in space. Snapshots of GCaMP6s fluorescence 28 min after posterior fold onset (right) show almost no activity in the anterior region compared to the control (left). (**G**) Disruption of calcium regulation in muscle cells inhibits constrictions. The probability of forming three folds is reduced under heat-shock induced expression of the dominant negative mutant allele *SERCA.R751Q* with a muscle-specific driver *Mef2-GAL4* ($N = 130$, $p = 7 \times 10^{-9}$), and is likewise reduced under RNA interference of MLCK driven by *tub67-GAL4;tub16-GAL4* ($N = 37$, $p = 2 \times 10^{-4}$). (**H**) Altogether, we infer that hox genes are upstream of patterned calcium pulses, which generate muscle contraction that is mechanically coupled to the endoderm, driving tissue strains and ultimately organ shape.

The online version of this article includes the following figure supplement(s) for figure 5:

**Figure supplement 1.** Kymographs of GCaMP6s dynamics show that calcium activity is initially localized in space to constrictions and begins near the time at which constrictions begin.

**Figure supplement 2.** *Antp* mutants show reduced calcium activity in the anterior two chambers for over an hour.

**Figure supplement 3.** Calcium pulses appear at the unaffected anterior constriction in *Ubx* mutants.

**Figure supplement 4.** Disrupting calcium activity hinders constrictions.

Importantly, we also find that in wild-type embryos, knock-downs of calcium signaling remove folds. In smooth muscle cells, calcium is known to trigger muscle contraction by binding to calmodulin, which in turn binds to myosin light chain kinase (MLCK) to trigger myosin light chain phosphorylation (*Hill-Eubanks et al., 2011*), and cytoplasmic calcium is transported from the cytosol into the sarcoplasmic reticulum for storage under regulation of SERCA (*Kuo and Ehrlich, 2015*). Driving a dominant negative form of SERCA previously shown to exhibit temperature-sensitive expression under *Mef2-GAL4* (*Kaneko et al., 2014*), we find that disrupting calcium signaling via heatshock suppressed midgut constrictions ($p = 7 \times 10^{-9}$, *Figure 5G*, *Video 9*, *Figure 5—figure supplement 4*). Separately, interrupting the production of MLCK in the muscle via RNA interference demonstrates a similar reduction in folding behavior ($p = 2 \times 10^{-4}$, *Figure 5G*). From this we infer that spatially localized calcium dynamics – under the control of hox gene patterning – triggers MLCK signaling leading to muscle contractions (*Figure 5H*).

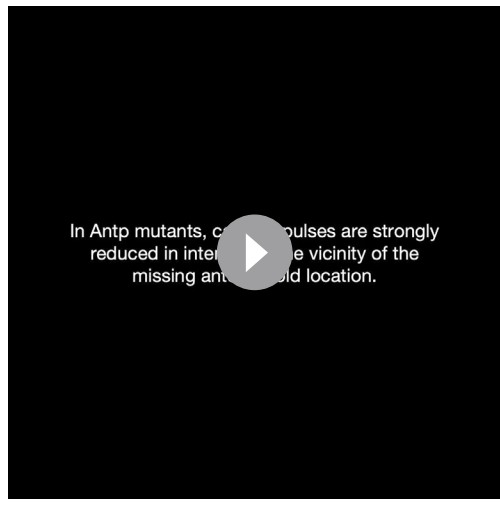

**Video 7.** Calcium pulses appear at the location of each constriction, shown here for the anterior constriction in a *Mef2-GAL4>UAS-GCaMP6s* embryo. We visualize the dynamics of calcium pulses by overlaying three snapshots captured nine seconds apart as red, green, and blue images for each frame. Thus, colored pixels represent transient activity reported by GCaMP6s. A composite frame is imaged every 90 s, and the timestamp is shown relative to the onset of the constriction, as monitored in a separate bright-field channel (not shown).

https://elifesciences.org/articles/77355/figures#video7

**Video 8.** In *Antp* mutants, calcium pulses are strongly reduced in intensity in the vicinity of the missing anterior constriction location. In WT embryos (top panel), calcium pulses appear at the onset of constrictions near the anterior constriction and appear in an increasingly spatially extended region as development progresses. In contrast, for *Antp* mutant embryos, calcium activity is reduced and does not exhibit an initially localized pattern (bottom panel).

https://elifesciences.org/articles/77355/figures#video8

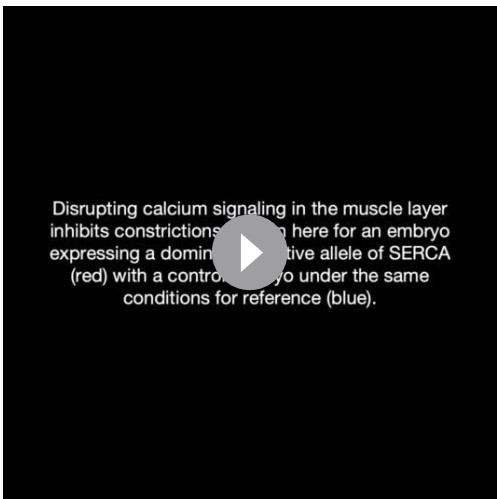

**Video 9.** Disrupting calcium signaling in the muscle layer inhibits constrictions, shown here for an embryo expressing a dominant negative allele of SERCA (red) with a control embryo under the same conditions for reference (blue).

https://elifesciences.org/articles/77355/figures#video9

## Discussion

Here, we studied morphogenesis of an organ in which heterologous tissue layers generate complex shape transformations. We found that convergent extension and sharp folds in the endodermal layer are triggered by mechanical interaction of muscle contractility together with tissue incompressibility, and patterns of calcium signaling regulate contractility in muscle cells according to hox-specified information (*Figure 5H*).

Although correspondences between hox genes and cell fates have been established for decades (*Bate and Martinez Arias, 1993*), understanding the physical processes driven by hox genes remains an active area of research. Here, we demonstrated a link from genes to tissue morpho-dynamics through active forces, connecting hox genes to a mechanical induction cascade across layers that integrates high-frequency calcium pulses to advance reproducible morphogenesis of complex 3D shape. While calcium dynamics have known roles in early developmental stages of diverse organisms (*Wallingford et al., 2001*; *Strutt, 2003*; *Li et al., 2019*; *Brodskiy et al., 2019*) – including influencing the organization of muscle fibers in the midgut (*Huycke et al., 2019*) and determining cell fates in heart valves (*Fukui et al., 2021*) – our findings suggest a direct influence of calcium on shape, wherein pulses trigger a program of irreversible tissue deformation. These calcium-patterned muscle contractions control 3D shape through a mechanical cascade across tissue layers, with broad relevance to tissue engineering and organ morphogenesis in other organisms.

At the cellular level, a remaining question is how the midgut selects precise positions and times for localized calcium activity despite broad hox gene domains that vary slowly with time. For example, the anterior fold forms near the center of the Antp domain. Do cells sense subtle gradients of Antp, or does more refined patterning downstream of hox gene expression specify this location (*Bilder and Scott, 1998*)? One available avenue for the latter possibility is that hox genes govern the formation of anatomical structures that may transmit signals from the soma to trigger calcium pulses (*Bataillé et al., 2015*).

At the tissue level, a remaining question is to what extent endodermal cells actively respond to muscle contraction, rather than passively deforming. For instance, there could be a mechanical signaling pathway provoking contractile behavior in endodermal cells, or even a mechanical induction loop between layers regulating morphological progression. Our findings open new avenues to study how dynamic interactions between layers encode complex shapes of visceral organs.

## Materials and methods
### Stocks and reagents

For this work, we used the following stocks based on Bloomington *Drosophila* Stock Center stocks: *w;48Y-GAL4;klar*, *w;UAS-CAAX::mCherry*, *w;UAS-histone::RFP*, *w;20xUAS-IVS-GCaMP6s*, *w;UAS-SERCA.R751Q.tdTomato;MKRS/TM6B Tb*, *UAS-MLCK RNAi TRiP VALIUM20*, *y,w,Antp-GAL4;klar*, *w;sqhGFP*, *w[*];I-76-D,Ubx⁹·²²,e¹/TM6B,Tb*, *Df(3 R)Antp^{NS+RC3},red¹,e¹/TM3,Sb,Ser*. Additional stocks used were *w;Mef2-GAL4* (a gift from Lucy O'Brien), *w[*];UASp-CIBN::pmGFP;UASp-mCherry::CRY2-OCRL* (a gift from Stefano de Renzis), *w[*];UASp-CIBN::pmGFP;UASp-RhoGEF2-CRY2::mCherry* (a gift from Stefano de Renzis), *Hand-GFP;4xUAS-Hand* (a gift from Zhe Han), *w;+;Ubx-GAL4 M1* (a gift from Ellie Heckscher) (*Garaulet et al., 2008*; *Crickmore et al., 2009*), *w;+;Ubx-GAL4 M3* (a gift from Ellie Heckscher) (*Garaulet et al., 2008*; *Crickmore et al., 2009*), *w;tub67-GAL4;tub15-GAL4* (a gift from Eric Wieschaus), and *w;+;Gap43 mCherry* (a gift from Eric Wieschaus).

We used following antibodies for staining: Ubx FP3.38 (diluted 1:10, Developmental Studies Hybridoma Bank), Antp 8C11 (diluted 1:25, Developmental Studies Hybridoma Bank), anti-ABDA C11 (dilued 1:50, Santa Cruz Biotechnology).

## Microscopy

For light sheet imaging of live and fixed embryos, we used a custom multi-view selective plane illumination microscope (confocal MuVi SPIM) drawn schematically in *Figure 1—figure supplement 1A*. This setup has been previously described in *Streichan et al., 2018*.

Subsequent registration, deconvolution, and fusion using the methods presented in *Preibisch et al., 2014* results in a single, deconvolved data volume per timepoint with isotropic resolution (0.018 µm$^3$ per voxel). For most experiments in this work, we acquire one volume per minute. The optimal number of deconvolution iterations varied between 8 and 20 for different fluorescent reporters. We used six or eight views for most datasets.

To peer deep inside the developing embryo, we leverage the UAS-GAL4 system (*Brand and Perrimon, 1993*) to express fluorescent proteins in gut-specific tissues and use embryos with the *klarsicht* mutation (*Welte et al., 1998*), which reduces scatter without altering gut morphogenesis. *Video 1* shows a maximum intensity projection of lateral, dorsal, and ventral half-data volumes for a deconvolved nuclear marker in the midgut as a complement to *Figure 1* of the main text. In *Video 1*, non-midgut tissues expressing *48Y-GAL4* are largely masked out using an iLastik training to darken voxels far from the gut, but some cells outside the midgut that express *48Y-GAL4* are visible, particularly on the dorsal side.

We used confocal microscopy (Leica SP8) for more detailed characterization of calcium dynamics, live imaging of *Ubx* and *Antp* mutants, and supplementary optogenetic experiments.

## Parameterization of the organ shape

To quantify the geometry of organ shape and deformation, we built an analysis package called TubULAR, as reported in *Mitchell and Cislo, 2022*. While details of the publicly-available, open source package are included in *Mitchell and Cislo, 2022* and the associated repository, here we describe our use of the package for the presented results.

We begin by using TubULAR's surface detection methods to extract the organ shape with a morphological snakes level set analysis (*Márquez-Neila et al., 2014*; *Chan and Vese, 2001*) on the output of an iLastik training (*Berg et al., 2019*) against midgut tissue (membrane, nuclei, actin, or myosin). Example results from this segmentation performed on a *w;48Y-GAL4/UAS-CAAX::mCh* embryo are shown in midsagittal sections in *Figure 1—figure supplement 2*. We then use TubULAR's cartographic mapping functions to map the surface to the plane and stabilize noise in the mappings' dynamics (*Mitchell and Cislo, 2022*). For visualization, we use a pullback parameterization $(s, \phi)$ such that the coordinate directions $(\hat{s}, \hat{\phi})$ are determined by the conformal mapping to the plane at the onset of the first constriction ($t = 0$). In this way, $\phi$ parameterizes the intrinsic circumferential axis and $s$ parameterizes a longitudinal position along the long axis of the organ at $t = 0$. In subsequent timepoints, the difference in parameterization coordinates in 3D space are minimized to match the previous timepoint, such that the coordinates follow the shape change of the organ (*Mitchell and Cislo, 2022*). We find this $(s, \phi, t)$ parameterization aids in both visualization and enables more accurate velocimetry measurements than other choices. We define the 'material frame' of a given midgut as the tissue configuration at the onset of the middle constriction, which is the first constriction that appears.

We compute the centerline using the TubULAR package (*Mitchell and Cislo, 2022*), wherein the organ is divided into circumferential 'hoops' based on its planar parameterization (*Figure 1—figure supplement 3*). Hoops for which s=constant define an effective circumference for increments along the length of the organ, and the average 3D position of each hoop defines its centerline point. Connecting mean points of adjacent hoops along the length of the organ defines the centerline of the object (brown curve) whose length is reported in the main text *Figure 1E*.

## Endodermal cell segmentation and shape change

Using a single slice of the gut surface projected into stabilized $(s, \phi)$ pullback coordinates, we segmented 600–1300 cells per timepoint (*Figure 2—figure supplement 1*) using a semi-automated procedure:

1. We first perform adaptive histogram equalization over patches of the pullback containing several cells in width.

2. We then perform two passes of morphological image reconstruction (see MATLAB's imreconstruct function) punctuated by morphological dilation and erosion steps.
3. The result is binarized and skeletonized via a watershed algorithm.
4. We overlay this skeleton on the original image to enhance the membrane contrast, convolve with a narrow Gaussian (with a standard deviation of $\sim 0.2\%$ of organ length), and pass the result through the previous three steps.

This gives us an estimate for the image segmentation. We then manually correct any spurious segmentation artifacts in GIMP (**The Gimp Development Team, 2019**) by overlaying the segmentation with the original pullback images. To resolve some ambiguous cell junctions, we examine not only a single slice of the endodermal cell layer near the apical side (about 2.5 µm beyond the apical side), but also the maximum intensity projection of several microns along the surface normal direction.

We compute cell anisotropy by finding segmented cell shapes in 2D, embedding those polygons in 3D, projecting each cell onto a local tangent plane, and measuring the moment of inertia tensor of this polygon in the material coordinate system. This procedure is shown schematically in **Figure 2—figure supplement 1A**. We then embed the Lagrangian coordinate directions $(\hat{s}, \hat{\phi})$ from a conformal mapping of the whole organ at the onset of the initial (middle) constriction $t = 0$ onto the cell's centroid in 3D (in the deformed configuration at time $t \neq 0$). The moment of inertia tensor for the cell polygon is expressed in the local coordinate system from the embedded $(\hat{s}, \hat{\phi})$ directions in the local tangent plane of the tissue. The eigenvalues $I_1$ and $I_2$ of the moment of inertia tensor and their associated eigenvectors then provide an effective ellipse for the cell with orientation θ with respect to the local $\hat{s}$ direction and an aspect ratio $a/b \equiv \sqrt{I_1/I_2}$. **Figure 2—figure supplement 1C** shows the raw data of these measurements without computing statistics.

We then average the cellular anisotropy over the organ surface to report a mean, standard deviation, and standard error for both the cellular aspect ratio and cell orientation. In this averaging, we weight each cell's contribution by its area, so that all material points on the organ are given equal weight. The results reported in **Figure 2C and D** in the main text show the weighted mean and weighted standard deviation for each distribution. The weighted means of the aspect ratio $a/b$ and orientation θ are

$$\langle a/b \rangle = \frac{\sum_{i=1}^{N} A_i a_i/b_i}{\sum_{i=1}^{N} A_i} \tag{2}$$

$$\langle \theta \rangle = \tan^{-1}\left[\frac{\sum_{i=1}^{N} A_i \sin\theta_i}{\sum_{i=1}^{N} A_i \cos\theta_i}\right], \tag{3}$$

where $A_i$ is the area of the $i^{\text{th}}$ cell, and $N$ is the total number of cells. We note that we obtain similar results by weighting each cell equally, which would correspond to setting $A_i = 1$ for all $i$ above.

We obtain standard errors by bootstrapping. In detail, we subsample our collection of measurements, compute the weighted mean for the subsample, and repeat with replacement 1000 times. The variance of these 1000 means decreases in proportion to the number of samples $n$ included in our subsampling, so that $\sigma_{\bar{x}}^2(n) = \tilde{\sigma}_{\bar{x}}^2/n + \sigma_0^2$. Fitting for $\sigma_{\bar{x}}^2$ across 50 values of $n$ in the range $N/4 < n < N$ and evaluating this fit for $n = N$ gives an estimate for the standard error on the mean $\sigma_{\bar{x}} = \sqrt{\sigma_{\bar{x}}^2}$. In practice, the result is nearly identical to measuring the means of many weighted subsamplings of $n = N$ cells with replacement and computing the standard deviation of this collection of means.

## Single-cell tracking

We tracked 175 cells from -27 min to 83 min of development relative to the onset of the middle constriction in a *w;48Y-GAL4;klar × w;UAS-CAAX::mCh* embryo imaged using confocal multi-view lightsheet microscopy (confocal MuVi SPIM). First, we segmented the same 175 cells in the first chamber of the gut every two minutes using the same procedure as in the previous section. We tracked their positions over time using the iLastik manual tracking workflow using 2D $(s, \phi)$ pullback projections. From these segmented polygons, we project back into 3D onto the gut surface and measure the cell areas in a local tangent plane for **Figure 2E** in the main text.

## Topological cell rearrangements in the endoderm

Cell rearrangements are also present in the endodermal tissue, and these 'T1' events could also contribute to the large-scale shear (*Etournay et al., 2015*). Importantly, the orientation of T1 transitions is not significantly aligned with the elongation axis at early times, suggesting that the endoderm is fluidized and that T1s are not a tightly controlled process directing morphogenesis.

To identify T1 transitions, we leveraged our single-cell tracking previously used in *Figure 2E* – a contiguous region of cells in the anterior chamber of the midgut, extending from the anteriormost portion of the midgut to the anterior fold. We query all cell pairs which share an edge in the endoderm at any time during the morphogenetic process. We then filter out any pairs that remain neighbors for all timepoints, since their shared edges do not participate in topological rearrangements. The remaining pairs reflect a cell-cell interface which either appears or vanishes during morphogenesis. We perform additionally screening of these candidate events to confirm that the change in cell topological is not an artifact from possible segmentation error by coloring the two cells participating and visually inspecting their motion. *Video 3* and *Figure 2—figure supplement 3* show an example sequence of T1 transitions tracked using this scheme.

For each junction lost or gained, we measure the axis of the associated T1 transition by computing the centroid of each cell in the pair that is gaining or losing a junction. *Figure 2—figure supplement 3C* shows a histogram of these axes' angles with respect to the anterior-posterior axis of the organ defined in a locally conformal coordinate patch, with the AP axis orientation inferred from the material (Lagrangian) frame. We find that T1 transitions oriented along the AP axis (converging along DV) occur about as frequently as T1 transitions oriented along the DV axis (converging along AP) for our collection of tracked cells in the first chamber, suggesting that topological cell rearrangements are not a principal driver of convergent extension in the tissue at early times. These rearrangements are therefore unlikely to drive shape change, as predicted by the quantitative similarity between tissue shear and cell shape change (*Figure 3—figure supplement 5*).

## Quantification of tissue deformation

To compute a coarse-grained tissue velocity over the gut surface, we again used the TubULAR package (*Mitchell and Cislo, 2022*). This resource enables velocimetry and discrete exterior calculus measurements (*Crane et al., 2013*) on the evolving surface. The result is a fully covariant measurement of the compressibility and shear of the tissue spanning the whole organ.

Briefly, given our $(s, \phi, t)$ coordinate system defined in the TubULAR pipeline, we then run particle image velocimetry (PIV) using PIVLab (*Thielicke and Sonntag, 2021*; *Thielicke and Stamhuis, 2014*) and map tissue velocities in the domain of parameterization to the embedding space. Geometrically, displacement vectors $\mathbf{v}$ extend from one $\mathbf{x}(s_0, \phi_0, t_0)$ coordinate in 3D on the surface at time $t_0$ to a different $\mathbf{x}(s_1, \phi_1, t_1)$ coordinate on the deformed surface at time $t_1$. When $t_0$ and $t_1$ are adjacent timepoints, this defines the 3D tissue velocity at $\mathbf{x}(s_0, \phi_0, t_0)$ as $\mathbf{v}(s_0, \phi_0, t_0) = (\mathbf{x}(s_1, \phi_1, t_1) - \mathbf{x}(s_0, \phi_0, t_0))/(t_1 - t_0)$. We decompose the velocity into a component tangential to the surface $\mathbf{v}_\parallel$ and a normal component $\mathbf{v}_n = v_n \hat{\mathbf{n}}$ for measuring divergence via discrete exterior calculus (*Mitchell and Cislo, 2022*) and for measuring out-of-plane deformation $2Hv_n$, where $H$ is the mean curvature obtained via computing the Laplacian of the mesh vertices in (embedding) space: $\Delta \mathbf{X} = 2H\hat{\mathbf{n}}$ (see *Crane et al., 2013*).

As shown in *Figure 3—figure supplement 1*, the in-plane dilatational flow almost perfectly matches the out-of-plane deformation during the morphogenetic process. We make sense of dilatational flow and out-of-plane deformation and interpret their difference as the local area growth rate by the following argument. The surface changes according to its tissue velocity, which has tangential and normal components $\mathbf{v} = \partial_t \mathbf{X} = v^i \mathbf{e}_i + v_n \hat{\mathbf{n}}$. The shape of the surface is encoded by the metric $g_{ij} = \partial_i \mathbf{X} \cdot \partial_j \mathbf{X}$, which describes lengths and angles measured in the tissue, and by the second fundamental form $b_{ij} = \partial_i \partial_j \mathbf{X} \cdot \hat{\mathbf{n}} = -\partial_i \mathbf{X} \cdot \partial_j \hat{\mathbf{n}}$, which contains information relating to both intrinsic and extrinsic measures of surface curvature (*Crane et al., 2013*). The time rate of change of the metric is determined by the superposition of velocity gradients and normal motion where the surface is curved (*Arroyo and Desimone, 2009*; *Marsden and Hughes, 1994*):

$$\partial_t g_{ij} = \nabla_i v_j + \nabla_j v_i - 2v_n b_{ij}. \tag{4}$$

Here, $\nabla$ denotes the covariant derivative operator defined with respect to the embedding metric $\mathbf{g}$. The covariant mass continuity equation gives (*Arroyo and Desimone, 2009*)

$$\begin{aligned} 0 &= \frac{D\rho}{Dt} + \frac{\rho}{2}\text{Tr}[\mathbf{g}^{-1}\dot{\mathbf{g}}] \\ &= \frac{D\rho}{Dt} + \rho\nabla\cdot\mathbf{v}_{\parallel} - \rho 2v_n H, \end{aligned} \tag{5}$$

where $\rho$ is the mass density in the physical embedding, and the material derivative is $D\rho/Dt = \partial_t\rho + \rho(\nabla\cdot\mathbf{v}_{\parallel}) + \mathbf{v}\cdot\nabla\rho$. Incompressibility ($D\rho/Dt = 0$) then implies

$$2Hv_n = \nabla\cdot\mathbf{v}_{\parallel}. \tag{6}$$

## Minimal ingredients demonstrate geometric interplay between compressibility and shear

A flat, nearly incompressible sheet demonstrates a kinematic coupling between dilatational in-plane flow ($\nabla\cdot\mathbf{v}_{\parallel}$) and out-of-plane deformation ($2Hv_n$). Contracting such a sheet as in *Figure 3—figure supplement 3A* leads to out-of-plane bending to preserve surface area of the sheet. This out-of-plane motion leaves cells unchanged in their aspect ratio: no in-plane deformation is necessary.

If the sheet is curved into a tube (so that mean curvature is nonzero, $|H| > 0$), then constricting an sheet (with inward velocity $v_n > 0$) can generate deformation in the local tangent plane of the sheet. Such incompressibility couples to initial curvature to generate shear deformation. For example, an incompressible sheet of paper glued into a cylinder along one of its edges cannot be squeezed in this fashion without crumpling, folding, or tearing. An elastic sheet, however, can be deformed in this manner even if local areas of material patches are required not to change. In particular, the sheet may stretch along the long axis while constricted circumferentially, such that a circular material patch is transformed into an elliptical patch with the same area, as in *Figure 3—figure supplement 3B*.

Finally, these two effects are coupled in the case of the pinched cylinder with a localized constriction. In a given snapshot with an existing localized constriction, we can schematically understand the three ingredients by considering the pinched cylinder with a step-wise indentation shown in *Figure 3—figure supplement 3C*. First, active stresses constrict the neck, decreasing the surface area of the neck (blue) and dilating the interior faces (red). In order to restore the surface area of the cells in the neck, its length may increase, resulting in extension along the long axis of the tube. In tandem, to combat the dilation in the interior faces, cells flow into the constriction from the chambers. Note that if all three steps are instantaneously coupled, the order of events is immaterial to the outcome: contractile surface flows could increase the density of cells in the interior faces, which leads to neck constriction to restore cell density in the faces and results in convergent extension of the neck.

We note also that when the constriction is broad along the longitude or when the indentation is shallow, the mean curvature will be positive everywhere (cylinder-like, $H > 0$). In this case, inward motion of the incompressible tube causes an extensile surface flow ($\nabla\cdot\mathbf{v}_{\parallel} > 0$), rather than a contractile one. This is a qualitative difference between broad or uniform constrictions of a tube and localized constrictions such as those seen in the midgut. In principle, we predict a crossover between the two modes of behavior during the very onset of constriction in our system – from positive to negative divergence as the curvature changes sign. This is a subtle and transient feature, given the large radius of the midgut compared to the small axial length of the constrictions and the non-uniform initial curvature of the midgut before constrictions begin.

## Quantification of tissue shear

We employ a geometric method of tissue-scale shear quantification that accounts for both the shear due to the changing shape of the gut and the shear due to the material flow of cells along the dynamic surface. The first step is to establish consistent material coordinates for all times – that is, labels for parcels of tissue that follow those parcels as they move and deform. We prescribe these labels at the onset of the first midgut constriction by endowing the organ's surface with a planar parameterization, as before. The cut mesh is first conformally mapped into a planar annular domain, $\{\|\mathbf{x}\| : r \leq \|\mathbf{x}\| \leq 1\}$, using a custom Ricci flow code included in the TubULAR package (*Mitchell and Cislo, 2022*). Fixing the outer radius of the annulus to 1, the inner radius $r$ is a conformal invariant that is automatically determined from the geometry of the organ. Taking the logarithm of these intermediate coordinates

then defines a rectangular domain, with a branch cut identifying the top and bottom horizontal edges of the domain in such a way the the cylindrical topology of the cut mesh in 3D is fully respected. The coordinates in this domain are taken to be the material ('Lagrangian') coordinate system, $(\tilde{\zeta}, \tilde{\phi})$. This conformal parameterization is, by construction, isotropic; the metric tensor is diagonal. This parameterization is therefore a suitable reference against which to measure all subsequent accumulation of anisotropy in the tissue. We note that at the reference time $t_0$ (at the onset of the first constriction), the material coordinates are similar to the $(s, \phi, t_0)$ coordinate system defined before, except that the coordinate $s$ measures a proper length on the surface along curves of constant $\phi$, while $\tilde{\zeta}$ is a coordinate of the conformally mapped planar domain. We chose to use $s$ instead of $\tilde{\zeta}$ for visualizations simply because deep constrictions exhibit extreme dilation in a conformal $(\tilde{\zeta}, \tilde{\phi})$ pullback plane, but these are attenuated in an $(s, \phi)$ pullback plane (see *Mitchell and Cislo, 2022*). We stress that all measurements account for the physical embedding of the surface: the coordinate system parameterizing the surface is a tool to define circumferential and longitudinal directions based on the organ's intrinsic geometry, and the choice of parameterization does not influence the magnitude of tissue deformation.

In order to recapitulate the material flow of the tissue, these coordinates are advected in the plane along the flow fields extracted using PIV (*Thielicke and Stamhuis, 2014*; *Thielicke and Sonntag, 2021*) and then mapped into 3D at each time point. This mapping defines a deformed mesh whose induced metric tensor, $\mathbf{g}' \equiv \mathbf{g}(t)$, can be computed relative to the material coordinates. All anisotropy in the mapping is encoded by the complex Beltrami coefficient, $\mu(t)$, defined in terms of the components of the time-dependent metric tensor

$$\mu(t) = \frac{g'_{11} - g'_{22} + 2i\, g'_{12}}{g'_{11} + g'_{22} + 2\sqrt{g'_{11}\, g'_{22} - g'_{12}{}^2}}. \tag{7}$$

As illustrated in *Figure 3—figure supplement 4B*, μ describes how an initially circular infinitesimal patch of tissue is deformed into an elliptical patch under the action of the material mapping. The argument of μ describes the orientation of this ellipse. The magnitude $|\mu|$ is related to the ratio $K$ of the lengths of the major axis of this ellipse to its minor axis by

$$K = \frac{1 + |\mu|}{1 - |\mu|}. \tag{8}$$

When $|\mu| = 0$, the material mapping is isotropic – that is, a circular patch of tissue remains circular under the mapping. Note that $|\mu| < 1$, and therefore μ provides a bounded description of both the magnitude and orientation of material anisotropy in the deforming surface.

The results of this measurement are shown as a kymograph in *Figure 3—figure supplement 4* for a representative dataset. Constrictions begin to appear at times and locations marked by red arrows and continue to deepen. The Beltrami coefficient is averaged along the circumferential direction and plotted at the anterior-posterior position in tissue coordinates at the onset of the middle constriction (the first constriction to appear), so that the deformation of advected tissue patches are compared to their original shape. A single color dominates the kymograph, indicating that the deformation is globally aligned to extend along the local longitudinal axis (and contract along the material frame's circumferential axis), despite the contorting and complex shape. This is consistent with circumferential muscle orientations defining the axes for convergent extension in the midgut.

## Relative motion between layers

To characterize relative motion between layers, we tracked 375 endodermal and 81 muscle nuclei in the same *w,Hand>GAL4;UAS-Hand:GFP;hist:GFP* embryo using a custom manual tracking workflow in MATLAB. *Figure 4—figure supplement 1* shows measurements of relative displacement of initially-close nuclei pairs (<5 μm apart at the onset of the first constriction). Two example tracks are highlighted in yellow and green. *Figure 4—figure supplement 2* shows additional statistics of the relative motion over time.

## Optogenetic experiments

The *UAS-CRY2-OCRL* and *UAS-RhoGEF2* constructs have been previously characterized (*Guglielmi et al., 2015*; *Izquierdo et al., 2018*). For optogenetic confocal microscopy experiments, we activated the optogenetic construct with continuous oblique illumination of a 470 nm LED at 6.2±0.1 mW/cm² power, in addition to periodic illumination with the 488 nm laser used to image the sample. Wild-type

embryos developed normally under this illumination ($N = 18/19$). Variations by a factor of two in either the LED power or in the 488 nm laser power used to image the *GFP* channel did not result in differences in phenotype. For light sheet imaging, we illuminated with a 488 nm laser line at 1 mW for 30 s once per minute.

We quantified the endoderm cell shapes using a similar procedure as before. After deconvolution (Huygens Essential software), we perform 3D segmentation via a morphological snakes level set method on an iLastik pixel classification to carve out an approximate midsurface of the endoderm. We measured the endoderm shape dynamics for two-color *y,w,Antp-GAL4;;Gap43 mCherry × w;UAS-CIBN::GFP;UAS-mCherry::CRY2-OCRL* embryos held under continuous optogenetic activation from oblique illumination of a 470 nm LED at 6.2±0.1 mW/cm² power as before. For comparison, we additionally measured endoderm shapes in *Antp* mutant embryos with a membrane marker driven in the midgut endoderm (*w,Antp^{NS+RC3};48Y-GAL4/UAS-CAAX::mCh*).

*Figure 4—figure supplement 3A* shows representative snapshots of this segmentation procedure for a two-color *y,w,Antp-GAL4;;Gap43 mCherry × w;UAS-CIBN::GFP;UAS-mCherry::CRY2-OCRL* embryo before the first constriction and 40 min after the middle constriction began. Circumferential muscle localized near the missing anterior constriction expresses the optogenetic construct (cyan band), while the endoderm is imaged using a ubiquitous membrane marker (grayscale). Image regions masked in semi-transparent gray are the deepest confocal plane acquired, while the rest of the image is a lateral view of the projected data on the segmented organ surface. Segmented endodermal cell polygons are colored by their aspect ratios. Cells are segmented in 2D and then projected into 3D for measurement of their aspect ratios. As shown in *Figure 4—figure supplement 3C*, there is no significant difference between cell orientations in wild-type (blue), optogenetic mutants (red), and *Antp* mutants (yellow).

## Wild-type calcium dynamics

We quantified calcium dynamics using confocal microscopy (Leica SP8) of the live reporter *UAS-GCaMP6s* driven by either the driver *Mef2-GAL4*, which is expressed across all muscles in the embryo, or *48Y-GAL4*, which is expressed in the embryonic midgut both in endoderm and visceral muscles. Here, we used *Mef2-GAL4* as a driver for characterizing anterior and middle constrictions. We used *48Y-GAL4* for the posterior constriction since many fluorescent somatic muscles occlude the line of sight for the posterior constriction under *Mef2-GAL4*, which complicated the analysis. We found that the two drivers yielded similar quantitative results for the anterior constriction.

To measure transient calcium activity without bias from variations in ambient fluorescent intensity due to spatially-dependent scattering, we imaged three confocal stacks with 2.5–3 μm step size in rapid succession (9 or 10 s apart) and subtracted subsequent image stacks from each other according to

$$\delta I \equiv |I_1 - I_2| + |I_2 - I_3| + |I_1 - I_3|, \tag{9}$$

where $I_i = I_i(x, y)$ is the maximum intensity projection (projected across approximately 30 μm) of the $i^{\text{th}}$ stack. Over such short timescales, motion of the midgut is small, but transient flashes of GCaMP6s are unlikely to span more than one acquisition. We then extract coherent features from $\delta I$ using a Gaussian blur followed by a tophat filter, and sum the resulting signal along the circumferential direction. This defines our measure of GCaMP activity. For *Figure 5B and C*, we report the raw background-subtracted signal for each frame and its integral over time.

While we interrogated GCaMP6s activity using many views of the gut, the quantification used three standardized views. For the anterior constriction, we used a dorsal view, since out-of-plane effects are smallest on the dorsal side and since the midgut is nearest to the surface on the dorsal side. For the middle constriction, we used ventrolateral views since there is a line of sight with fewer other muscles driven by *Mef2-GAL4* from this view. For the posterior constriction, we used a left lateral view for quantification.

To time-align the GCaMP experiments of the anterior and middle constrictions, we defined $t = 0$ as the first timestamp in which the constriction under observation showed localized bending along the longitudinal (AP) axis. For characterization of calcium dynamics at the posterior constriction, we defined the onset of constriction by the ventral side of the gut visceral muscle having moved dorsally by ~10 μm.

*Figure 5—figure supplement 1* shows kymographs of GCaMP6s dynamics averaged across biological repeats. In these kymographs, activity begins near the time when constrictions begin.

## Calcium activity in *Antp* mutants

*Figure 5—figure supplement 2* shows delayed and suppressed GCaMP activity in *Antp* mutants compared to the wild-type behavior of sibling embryos that are not homozygous mutants for *Antp*. To compare calcium activity in *Antp* mutants against wild-type dynamics, we computed p values using a *z*-score measuring the difference between *Antp* heterozygotes (controls) and *Antp* homozygotes (mutants) as

$$Z = \frac{\overline{\delta I}_{\text{mutant}} - \overline{\delta I}_{\text{control}}}{\sqrt{s_{\text{control}}^2/n_{\text{control}} + s_{\text{mutant}}^2/n_{\text{mutant}}}}, \tag{10}$$

where $\overline{\delta I}$ is the sample mean, $s_{\text{control}}$ and $s_{\text{mutant}}$ are the sample standard deviations, and $n_{\text{control}}$ and $n_{\text{mutant}}$ are the sample sizes. This score gives a single-sided p value via

$$p = \tfrac{1}{2}\text{erfc}\left(-Z/\sqrt{2}\right), \tag{11}$$

where erfc is the complementary error function.

To quantify the difference in overall activity between mutants and heterozygotes, we first estimate the expected fluorescent intensity for a given embryo under the null hypothesis that all embryos, whether mutant or not, will have similar GCaMP6s activity. Since embryos vary in opacity, we normalized each heterozygous embryo according to a value dependent on its background fluorescent intensity measured in regions within the embryo but far (45–50 μm) from the site of the putative constriction. The observed maximum fluorescent activity $\delta I$ correlated with this background signal with a correlation coefficient of 78% and a mean signal-to-background ratio of 5.1±0.5. We then normalized each embryo's time-averaged $\delta I = \delta I(x)$ as

$$\delta I \to \frac{\delta I - \delta I_{\text{bg}}}{\delta I_{\text{max}} - \delta I_{\text{bg}}}. \tag{12}$$

This enabled us to reduce the confounding influence of variation in optical density between embryos in the mutant analysis, so we could compare absolute $\delta I$ curves rather than only their variation along the anterior-posterior axis.

## MLCK RNAi and SERCA mutant analysis

To drive expression of MLCK RNAi or a dominant negative allele of *SERCA*, we administered heat shock by abruptly raising the temperature to 37° C using a stage-top incubator (Okolab) and observing embryos staged such that they had not yet completed gut closure at the initiation of heatshock. The standard errors in the probabilities of successful constrictions are given by

$$SE = \sqrt{\frac{\hat{p}\left(1 - \hat{p}\right)}{N}}, \tag{13}$$

where $\hat{p}$ is the observed frequency of forming all three constrictions and $N$ is the number of samples of a given genotype (for ex, *Mef2-GAL4 × UAS-SERCA.R751Q.tdTomato*) measured in the experimental heat shock conditions. We note that the result is not sensitive to the details of the analysis. For example, we also computed the mean number of folds formed – that is, the number of deep constrictions – for each condition and compare the two distributions, as shown in *Figure 5—figure supplement 4A and B*. The mean number of folds formed was reduced in both *Mef2-GAL4 × UAS-SERCA.R751Q* embryos and *tub67-GAL4;tub16-GAL4 × UAS MLCK RNAi* embryos ($p = 3 \times 10^{-8}$ and $p = 0.002$, respectively).

## Acknowledgements

This research was supported by NIH Grant No. R35 GM138203 and NIH Grant No. R00 HD088708. We thank members of the Streichan and Shraiman labs, Eric Wieschaus, and Zvonimir Dogic for valuable insights, discussions, and suggestions. Isaac Breinyn aided in early exploration of the system. Matt Lefebvre, Isaac Breinyn, and Sophie Streichan aided in handling several stocks, crosses, and reagents. We acknowledge Ben Lopez in the NRI-MCDB Microscopy Facility for support and maintenance of the Resonant Scanning Confocal supported by the NSF MRI grant DBI-1625770. The authors acknowledge the use of the Microfluidics Laboratory (Innovation Workshop) within the California NanoSystems Institute, supported by the University of California, Santa Barbara and the University of

California, Office of the President. NPM acknowledges support from the Helen Hay Whitney Foundation. SS acknowledges support from the Harvard Society of Fellows.

# Additional information

### Funding

| Funder | Grant reference number | Author |
|---|---|---|
| National Institutes of Health | R35 GM138203 | Sebastian J Streichan |
| National Institutes of Health | R00 294 HD088708 | Sebastian J Streichan |
| Helen Hay Whitney Foundation | F-1246 | Noah P Mitchell |
| National Science Foundation | PHY-1748958 | Boris I Shraiman |
| Harvard University | Harvard Society of Fellows | Suraj Shankar |

The funders had no role in study design, data collection and interpretation, or the decision to submit the work for publication.

### Author contributions

Noah P Mitchell, Data curation, Formal analysis, Methodology, Software, Visualization, Writing – original draft, Writing – review and editing, Investigation; Dillon J Cislo, Formal analysis, Software, Visualization; Suraj Shankar, Boris I Shraiman, Formal analysis; Yuzheng Lin, Formal analysis, Investigation; Sebastian J Streichan, Conceptualization, Formal analysis, Funding acquisition, Investigation, Methodology, Project administration, Supervision, Writing – original draft, Writing – review and editing

### Author ORCIDs

Noah P Mitchell ![ORCID] http://orcid.org/0000-0003-1922-8470
Suraj Shankar ![ORCID] http://orcid.org/0000-0002-4615-975X
Boris I Shraiman ![ORCID] http://orcid.org/0000-0003-0886-8990
Sebastian J Streichan ![ORCID] http://orcid.org/0000-0002-6105-9087

### Decision letter and Author response

Decision letter https://doi.org/10.7554/eLife.77355.sa1
Author response https://doi.org/10.7554/eLife.77355.sa2

# Additional files

### Supplementary files

• MDAR checklist

### Data availability

We have uploaded processed data for experiments spanning all figures to FigShare, available at https://doi.org/10.6084/m9.figshare.c.6061508.v2. The original volumetric data from living imaging are each up to a terabyte in size. We therefore posted processed data on FigShare, including 2D pullback image sequences of the dynamic 3D tissue surfaces, volumetric data for small datasets, and processed tables. An interested researcher would be able to access the original data on our lab server. They would need to contact Sebastian Streichan (streicha@physics.ucsb.edu) to be added to the server's list of users and could then download the original data directly. In addition to the TubULAR package detailed in Mitchell & Cislo, 2022, further software and scripts used to analyze the data is available at: https://github.com/npmitchell/VisceralOrganMorphogenesisViaCalciumPatternedMuscleConstrictions, (copy archived at swh:1:rev:3835e0a1faa5853939c2967c867dde5cf49d9887).

The following dataset was generated:

| Author(s) | Year | Dataset title | Dataset URL | Database and Identifier |
|-----------|------|---------------|-------------|-------------------------|
| Mitchell N | 2022 | Data for 'Visceral organ morphogenesis via calcium-patterned muscle constrictions' | https://doi.org/10.6084/m9.figshare.c.6061508.v2 | figshare, 10.6084/m9.figshare.c.6061508.v2 |

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
