## [Editor Report]

Using state-of-the-art light sheet microscopy and novel image analyses, this study shows how genetically encoded contractions of the visceral muscle shape morphogenesis of the *Drosophila* midgut via mechanical coupling. Muscle contractions induce cell shape changes that cause convergence and extension of the gut epithelium, which folds due to its incompressible nature. Thus, this work defines novel mechanisms of convergence and extension and tissue folding via the interaction of a genetic program with physical processes.

---

## [Decision Letter]

**Decision letter after peer review:**

Thank you for submitting your article "Visceral organ morphogenesis via calcium-patterned muscle constrictions" for consideration by *eLife*. Your article has been reviewed by 3 peer reviewers, and the evaluation has been overseen by a Reviewing Editor and Didier Stainier as the Senior Editor. The reviewers have opted to remain anonymous.

All reviewers agree this is a highly interesting and technically sophisticated study that represents a significant advance for the field both conceptually and technically. While the conclusions are well supported by experimental data, the reviewers point out several points that need to be elaborated further with text changes or that need to be discussed. They also pointed out one instance in which analysis of existing data could improve the manuscript and made some suggestions that the authors should consider as well. Also included in the critiques below is a list detailing minor corrections that need attention.

Essential revisions:

Analysis of existing data:

1. The images in Figure 2B show that cells in the future-fold region had a more elongated shape than cells in the adjacent region at the initial developmental stage. I wonder if there is a correlation between cellular deformation and the determination of the fold position. This can be addressed by analyzing cells in the future-fold region and adjacent region separately (Figure 2C-E). Figure 2F may be modified or removed if spatial inhomogeneity in the cellular deformation is detected by the analysis suggested above.

Discussion points:

2. The authors demonstrated that dilatational flows are accommodated with out-of-plane deformation to preserve the cell areas. However, it is not a priori clear that such local constrictions lead to convergent extension. In principle, the local constrictions and dilatational flows could also lead to the shortening of the tube. Authors should provide additional explanations as to what causes the convergent extension.

3. Endodermal cells are shown to change shape and flow, but do they exchange neighbors? The contribution of (or lack of) neighbor exchanges to endodermal shape change should be discussed as this is the primary mechanism by which we know epithelial tissues to converge and extend. Specifically, In Line 169: "As shown in Figure 3H-I, cell segmentation of the endoderm during optogenetic inhibition of muscle contraction in the Antp domain reveals nearly constant aspect ratios: the endoderm cells near the Antp domain undergo reduced convergent extension when muscle contraction is locally disrupted." Are the authors equating cell aspect ratio change as a change in convergent extension? Providing clear definitions for cell and tissue-scale behaviors in the main text would be helpful.

4. In Eq. (6) the out-of-plane deformation is related to the divergence of the total velocity. However, for the data analysis authors equated the out-of-plane deformation to the divergence of the tangent component of the velocity. What is the reason for this difference?

5. The novel computational framework called 'TubULAR' in Ref. 6 of the supporting information is not publicly available. Details of the computational framework should either be posted on the public preprint or included in this manuscript.

Suggestions:

6. The manuscript is very compact, and many aspects and methods are not fully explained in the main text, please consider decompressing the narrative and moving some of the key methods and associated figures to the main text.

7. Quantifying the cellular deformation and Ca++ spikes in unaffected folds in mutant embryos and/or optogenetically manipulated embryos will further support the claim that Hox codes spatially restrict muscle constriction and resultant endodermal cell shape changes, leading to folding of the midgut.

---

## [Author Response]

Essential revisions:Analysis of existing data:1. The images in Figure 2B show that cells in the future-fold region had a more elongated shape than cells in the adjacent region at the initial developmental stage. I wonder if there is a correlation between cellular deformation and the determination of the fold position. This can be addressed by analyzing cells in the future-fold region and adjacent region separately (Figure 2C-E). Figure 2F may be modified or removed if spatial inhomogeneity in the cellular deformation is detected by the analysis suggested above.

Thank you for highlighting this feature. Indeed, we do measure a peak in the endodermal cell anisotropy near the anterior constriction and middle constriction precursor locations, but no peak in endodermal cell anisotropy exists at the posterior constriction precursor region. We discuss this point in the main text, plotting this spatial dependence of endodermal cell aspect ratios in Figure 2 – supplement 2. In addition, we note that the correlation between *tissue* deformation and the determination of the fold position was originally mentioned only briefly and now is addressed in more detail.

Figure 2F (which now corresponds to Figure 3E) was meant to *contrast* the behavior of a uniformly convergent-extending tube with the *localized* constrictions of the gut. The reviewers’ comment demonstrates that was unclear. We now have designed Figure panels 3D-F and expanded text to explain this point.

Discussion points:2. The authors demonstrated that dilatational flows are accommodated with out-of-plane deformation to preserve the cell areas. However, it is not a priori clear that such local constrictions lead to convergent extension. In principle, the local constrictions and dilatational flows could also lead to the shortening of the tube. Authors should provide additional explanations as to what causes the convergent extension.We agree that it is not a priori clear that local constrictions lead to convergent extension, and we drew this connection – and the connection between outof-plane constrictions and associated in-plane flows – only after careful study. Because this kinematic process is inherently 3D, the mechanism of generating convergent extension via constrictions can be difficult to conceptualize succinctly. We have opted to decompress this section in order to explain the kinematics in more detail. In addition to revision in the main text, Figure 3 D-F and a new supplementary figure further clarify this point.

Importantly, is not merely the projected length of the organ that is undergoing extension, but longitudinal geodesics in the deforming tissue (i.e. curves along the AP axis like the orange curve in Figure 3C). To the referee’s point of tube shortening, there indeed could be a shortening of the tube along the AP axis in the Eulerian frame (in the embedding space) while the longitudinal length along the bending tissue increases near constrictions. In fact, the projected length of the gut in the embedding space does not shorten in the embedding space, but lengthens, albeit not as dramatically as the centerline length. This is evident in Figure 1 – supplements 2 and 3 and in Videos 1 and 2. As the constriction narrows, the constrictions’ circumference decreases, while the deforming longitudinal axis extends, as highlighted in Figure 3C. We have expanded and clarified the section discussing tissue deformation in the main text, adding more detail to avoid misunderstandings.

Specifically, we split Figure 2 into two figures, added several panels, and split the discussion of tissue kinematics into two sections. The first section focuses on cell-scale deformations, while the second focuses on how these cell-scale deformations integrate into tissue-scale shape change.

3. Endodermal cells are shown to change shape and flow, but do they exchange neighbors? The contribution of (or lack of) neighbor exchanges to endodermal shape change should be discussed as this is the primary mechanism by which we know epithelial tissues to converge and extend. Specifically, In Line 169: "As shown in Figure 3H-I, cell segmentation of the endoderm during optogenetic inhibition of muscle contraction in the Antp domain reveals nearly constant aspect ratios: the endoderm cells near the Antp domain undergo reduced convergent extension when muscle contraction is locally disrupted." Are the authors equating cell aspect ratio change as a change in convergent extension? Providing clear definitions for cell and tissue-scale behaviors in the main text would be helpful.

This is an important question. As now detailed in our revised manuscript, we find that tissue-scale convergent extension is accounted for primarily by cell shape change at early times. We also include a new supplementary movie demonstrating topological rearrangements in the endoderm.

As the reviewers rightly note, convergent extension can be driven by multiple processes, including oriented cell divisions and cell intercalations. However, we find no signs of cell division in the midgut throughout morphogenesis, ruling out oriented divisions. While cell rearrangements are present in the endodermal tissue (Video 3 and Figure 2 – supplement 3), tracking neighbor exchanges in the endoderm of the anterior midgut reveals that rearrangements are not aligned in their orientation during the early stages of constrictions. Our measurements suggest these cell rearrangements are not a tightly controlled process directing morphogenesis, and instead reflect fluidized behavior of the anisotropic endoderm. Without divisions or oriented cell intercalations, we asked whether cell shape change alone can explain the tissue scale convergent extension. Figure 3 supplement 5 shows a quantitative match between cell shape changes and tissue convergent extension, indicating that local cell shape changes primarily mediate meso-scale convergent extension.

4. In Equation (5) the out-of-plane deformation is related to the divergence of the total velocity. However, for the data analysis authors equated the out-of-plane deformation to the divergence of the tangent component of the velocity. What is the reason for this difference?

The reviewers astutely recognize that we dropped the subscript ∥ in equation (5) of the Methods (now equation 6). We define out-of-plane deformation as 2*Hv_n_*, where *v_n_* is the out-of-plane velocity and *H* is the mean curvature. In an incompressible tissue, this quantity is equal to the in-plane divergence of the in-plane velocity, ∇ · **v**_∥_, so that

2*Hv_n_* ≈ ∇ · **v**_∥_*.* (1)

We ensure now that all in-plane velocity terms carry this subscript. Moreover, we have expanded the tissue deformation section of the main text to be clearer about the definitions of the quantities we discuss and how they relate to each other.

We note that tissue kinematics describes how in-plane and out-of-plane deformations are related, but does not distinguish causality between the two quantities. As discussed in the revised text, the kinematic constraint of incompressibility guides the shape changes that result from prescribed patterns of mechanical stresses in the tissue.

5. The novel computational framework called 'TubULAR' in Ref. 6 of the supporting information is not publicly available. Details of the computational framework should either be posted on the public preprint or included in this manuscript.

We have submitted a preprint detailing TubULAR on bioRxiv (reference 32 of this manuscript, doi.org/10.1101/2022.04.19.488840) and the references to the corresponding GitHub documentation therein. In the present work, we leverage this tool for addressing aspects of organ morphogenesis shown in Figures 1-3. The authors are in the process of revising this preprint into a paper showcasing the toolkit’s applicability to different biological systems, to be submitted for in-depth review with a journal. Detailing this toolkit’s functionality would be beyond this scope of the present manuscript.

Suggestions:6. The manuscript is very compact, and many aspects and methods are not fully explained in the main text, please consider decompressing the narrative and moving some of the key methods and associated figures to the main text.

We have followed this suggestion by expanding the main text and re-formatting the key methods figures as main text figure supplements. We have also split Figure 2 into two figures and expanded our discussions.

7. Quantifying the cellular deformation and Ca++ spikes in unaffected folds in mutant embryos and/or optogenetically manipulated embryos will further support the claim that Hox codes spatially restrict muscle constriction and resultant endodermal cell shape changes, leading to folding of the midgut.

We have measured calcium dynamics in the unaffected anterior constriction of Ubx mutants and find pulsatile activity localized to the anterior constriction, consistent with WT behavior for that constriction (see Figure 5 – supplement 4). This suggests that the calcium dynamics at each constriction are regulated locally along the midgut.

We would like to further address the dynamics of muscle cell deformation during Calcium spikes on the rapid timescales at which pulses occur. Simultaneous, rapid imaging of muscle cell deformation and/or endodermal cell deformation with a calcium reporter in the muscle layer will enable a quantitative correlation. Given the 3D shape of the long muscle cells and the rapid timescales at which pulses appear and decay, this presents some imaging challenges which we are currently working to address in future work.